# Asymmetric Duos: Sidekicks Improve Uncertainty

**Tim G. Zhou**[1,2][*]  **Evan Shelhamer**[1,2]  **Geoff Pleiss**[1,2]

[1]University of British Columbia  [2]Vector Institute

## Abstract

The go-to strategy to apply deep networks in settings where uncertainty informs decisions—ensembling multiple training runs with random initializations—is ill-suited for the extremely large-scale models and practical fine-tuning workflows of today. We introduce a new cost-effective strategy for improving the uncertainty quantification and downstream decisions of a large model (e.g. a fine-tuned ViT-B): coupling it with a less accurate but much smaller "sidekick" (e.g. a fine-tuned ResNet-34) with a fraction of the computational cost. We propose aggregating the predictions of this *Asymmetric Duo* by simple learned weighted averaging. Surprisingly, despite their inherent asymmetry, the sidekick model almost never harms the performance of the larger model. In fact, across five image classification benchmarks and a variety of model architectures and training schemes (including soups), Asymmetric Duos significantly improve accuracy, uncertainty quantification, and selective classification metrics with only $\sim 10 - 20\%$ more computation. Code is available at: https://github.com/timgzhou/asymmetric-duos.

## 1  Introduction

Deep ensembles [29] have endured as a de facto method in settings that require uncertainty quantification (UQ) for safety or downstream decision making [4, 9, 31]. When models are trained from scratch with different random seeds, this simple strategy yields strong performance across UQ and decision-making benchmarks [22, 39], provides robustness under distribution shifts [42], and requires no modification to model training methodology.

However, as the modern machine learning paradigm increasingly revolves around larger and larger pre-trained models fine-tuned for specific downstream tasks [12, e.g.], deep ensembles become increasingly ineffective and impractical. First, unlike training models from scratch, fine-tuning from a pre-trained model is much less sensitive to hyperparameters and training randomness. While ensembling multiple fine-tuning runs can yield performance increases, it pales in comparison to the gains obtained when ensembling trained-from-scratch models [19, 49]. Second, and more critically, the growing size of modern models makes ensembling more expensive than ever. Even with moderate-sized models like ResNet-50 [24], training and inference costs were a concern [1, 13, 26]; with large architectures such as ViT-H [10], these costs become even more prohibitive. Deploying a single one of these models can approach the limit of practical resources. Ensembling with double, triple, or larger multiples of the computation may be out of the question.

The smallest deep ensemble requires two models, which with standard approaches needs $2\times$ the computation for both training and inference. In this paper, we reduce this cost to a fractional amount (e.g. $10 - 20\%$) while still obtaining meaningful improvements in accuracy, uncertainty quantification, and downstream decision making. Our *Asymmetric Duo* framework pairs a base model with a smaller "sidekick" to make better predictions together. For instance, we can pair a larger ViT-B with a smaller ResNet-34, while fine-tuning each separately, to improve *over either alone*. As the

---

[*]tgzhou@cs.ubc.ca

39th Conference on Neural Information Processing Systems (NeurIPS 2025).

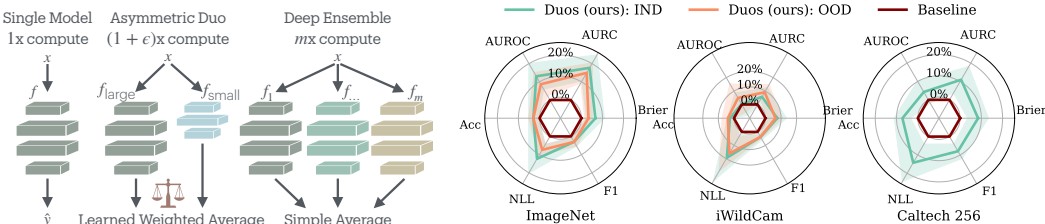

Figure 1: **Overview of Asymmetric Duos.** (Left) Schematic of Duos vs. single models and deep ensembles. (Right) Gains from Duos where the sidekick adds only 10%–20% more FLOPs depending on the choice of models. Asymmetric Duos improve upon their base models across accuracy, uncertainty quantification, and selective classification metrics on in-distribution (IND) and out-of-distribution (OOD) data (see Sec. 5 for experiment details). We plot relative improvements toward perfect scores; shading marks the standard deviation across the choice of base and sidekick models.

sidekick model requires a fraction of the compute at training and inference time, it only requires minimal computational overhead over the base model, which enables practical use.

Intuitively, an Asymmetric Duo might not help or might even hurt. The smaller model's accuracy can be much lower, and so one might expect that combining predictions from both models could degrade performance. Yet surprisingly, we find that Asymmetric Duos consistently improve performance, even when the base model and sidekick are extremely imbalanced (i.e. the sidekick is $10\times$ smaller and $80\%$ as accurate as the base model). Across datasets and model families, Asymmetric Duos yield up to $10-15\%$ improvements in accuracy, uncertainty quantification, and selective classification metrics, while only adding $10-20\%$ of the computational cost of the base model.

The Asymmetric Duo framework is straightforward to implement: given a working base model, the user adds a second smaller model (by say simply downloading it from a "model zoo") and fine-tuning it in the same way as the base model. Moreover, Asymmetric Duos are complementary to other compute-efficient model combinations, like Model Soups [60], and apply broadly by incorporating these methods as base or sidekick models. Our experiments show that Duos can improve the uncertainty quantification of both uncertainty-blind pipelines (like standard training) and uncertainty-aware pipelines (like soups), and so offer a practical drop-in addition to existing models.

## 2 Background

We focus on deep networks fine-tuned for a $k$-way classification task in some input domain $\mathcal{X}$. However, the key ideas of our Duos could be extended to regression tasks or other problem settings.

We denote deep networks as mappings of the form $f : \mathcal{X} \to \mathbb{R}^k$, where output logits $f(X)$ are mapped to the $k$-class probability simplex via the softmax function $\sigma(Z) = [e^{Z_i} / \sum_{j=1}^{k} e^{Z_j}]_{i=1}^k$. The deep network's class prediction for an input $X$ is the index of the largest logit (i.e. $\hat{Y}(X) = \arg\max_i [f(X)]_i$). Its predictive uncertainty can be quantified via its associated softmax probability:

$$\text{unc}(f(X)) = 1 - [\sigma(f(X))]_{\hat{Y}(X)}. \tag{1}$$

(We note that alternative measures of predictive uncertainty, such as the entropy of the categorical distribution defined by $\sigma(f(X))$, are common in other works [2, 29, 57]. We explore entropy uncertainty in Appendix C and find no meaningful difference with respect to our main results.)

### 2.1 Deep Ensembles

A deep ensemble [29] refers to $m$ independently-trained deep networks $f_1, \ldots, f_m$, typically of identical architecture and training procedure. At test time, each input $X$ is passed through each network and the resulting logits are averaged to produce an aggregate prediction $\bar{f}(X) := \frac{1}{m} \sum_{i=1}^{m} f_i(X)$ from which a class prediction and uncertainty estimate can be derived. The randomness of initialization and shuffling of training samples is usually sufficient to produce diverse logits that, when averaged together, produce more accurate class predictions and better calibrated

uncertainty estimates than any of the component models in isolation [29, 42]. Deep ensembles have also been shown to be robust to distribution shifts [2, 32, 42] and have thus been adopted across safety-critical and decision-making applications [4, 9, 31].

As stated in the introduction, deep ensembles are often impractical with the modern transfer learning setup [41, 49, 60]. The primary downside is computational cost. It is common for practitioners to fine-tune large pre-trained models, typically downloaded from a "model zoo", and training or deploying multiple of these large models can be prohibitively expensive [7]. In addition, member models fine-tuned from a shared pre-trained model tend to settle within the same loss basin, preventing the necessary diversity to produce meaningful improvements [40]. Consequently, deep ensembles are often restricted to settings with moderate-sized models trained from scratch [29, 42, 49].

## 2.2 Alternative Methods

Many methods aim to replicate the benefits of ensembles without the computational overhead.

**Model soups** [60] average the parameters of independently fine-tuned models, rather than their predictions, yielding a "weight-space" ensemble with the same inference cost as a single model. This method takes advantage of the fact that models fine-tuned from an identical base model often settle into the same pre-trained loss basin, and thus the loss is approximately locally convex with respect to the parameters. This parameter averaging often yields competitive performance with the prediction averaging of ensembling when the component models are fine-tuned, though the gain is less than ensembling trained-from-scratch models.

**Shallow ensembles** [30] reduce the multiple model training and inference costs of deep ensembles by instead making multiple predictions from a single model. Specifically, the last layer (or the last $\ell$ layers) of a deep network are replaced with $m$ copies that each receive the same input. The outputs of the $m$ "heads" are averaged together and learned jointly. Since only the last layer is duplicated, the computation and parameters added by a shallow ensemble are only a fraction of that of the original deep network. However, the benefits provided by shallow ensembles are weaker and less consistent than those of deep ensembles. In fact, single model baselines sometimes outperform them on selective classification tasks and robustness under distribution shifts [39].

**Other methods.** *Implicit ensembles* aim to produce multiple diverse networks at inference that are all derived from a single training run. These networks are derived through Monte Carlo sampling (e.g. when combined with Dropout [13]), checkpoints from the training process [26], or predictions made at early layers [1]. While these approaches have lower training costs than standard deep ensembles, they retain the same inference costs and typically are incompatible with fine-tuning. *Weight-sharing ensembles* aim to embed multiple models within a single architecture that can be trained simultaneously [11, 55, 58, e.g.]. While some methods incur lower inference-time costs [23], they typically require from-scratch training and bespoke architectures and/or training techniques. *Non-ensemble* methods often modify architectures, training, or inference; while they can be effective, these modifications may complicate practical use. See [15, 44] for thorough surveys.

## 3 Asymmetric Duos

We introduce *Asymmetric Duos* to augment a large base model $f_{\text{large}}$ with a small sidekick model $f_{\text{small}}$, which are both independently pre-trained and fine-tuned, for improving inference at little computational cost. Duos allow many choices of base $f_{\text{large}}$ and sidekick $f_{\text{small}}$. It is practical to choose a base that could be deployed by itself for adequate predictive performance and aggregate it with a much smaller sidekick, even if the sidekick cannot satisfactorily address the task on its own.

We contrast our pairing of asymmetric models with deep ensembles. Even the smallest deep ensemble with $m = 2$ models doubles the training and inference cost. To sidestep this cost, we instead look to combining models of different sizes. At first glance, this strategy may appear to have limits; too much imbalance may produce worse predictions than the base model alone. However, through a carefully designed aggregation strategy, we show that our Asymmetric Duos can improve performance even when the $f_{\text{small}}$ uses an order of magnitude less computation than $f_{\text{large}}$.

The Asymmetric Duo framework is architecture-agnostic and compatible with current transfer-learning and regularization techniques. This flexibility offers two key advantages. First, Duos can

more readily adjust the amount of computation, compared to standard ensembling, by extending to combinations of models with vastly different sizes, which is relevant given the wide span of pre-trained models available today. Second, Duos only add a small sidekick model to a larger base model, and so the additional computation is merely a fraction of that of the larger model.

## 3.1 Aggregating Predictions from Asymmetric Models

In standard deep ensembles, all models are inherently "equal" in that they only differ in randomization. By contrast, the two networks in an Asymmetric Duo explicitly differ in size and predictive performance. For example, an Asymmetric Duo might combine a ConvNeXt-Base (89 million parameters, 15 billion FLOPs, 84.1% Top-1 accuracy on ImageNet1K) [35] and a RegNeXt-1.6GF (9 million parameters, 1.6 billion FLOPs, 79.7% Top-1 accuracy on ImageNet1K) [46]. While this asymmetry achieves computational efficiency, the larger model is more likely to be correct than the other. Aggregating predictions with the equal-weighted logit average of deep ensembles could therefore be detrimental.

While many mechanisms could account for the asymmetry between $f_{\text{small}}$ and $f_{\text{large}}$ when aggregating predictions, we propose a simple weighted average of their logits. Specifically, we include two temperature parameters, $T_{\text{large}}$ and $T_{\text{small}}$ to weight the logits:

$$f_{\text{Duo}}(X) = f_{\text{large}}(X) \cdot T_{\text{large}} + f_{\text{small}}(X) \cdot T_{\text{small}}.$$

The Duo class predictions and uncertainty measure are based on these weighted logit averages:

$$\hat{Y}_{\text{Duo}}(X) = \arg\max_i [f_{\text{Duo}}(X)]_i, \qquad \text{unc}(f_{\text{Duo}}(X)) = 1 - [\sigma(f_{\text{Duo}}(X))]_{\hat{Y}_{\text{Duo}}(X)}.$$

**Tuning the temperatures.** We tune $T_{\text{large}}$ and $T_{\text{small}}$ to minimize the negative log likelihood on the same set used for hyperparameter selection during fine-tuning. As there are only two parameters, the learned weights will likely generalize even from a small reused validation set. With $f_{\text{large}}$ and $f_{\text{small}}$ validation outputs saved, this tuning takes only seconds with any standard optimizer. Note that this procedure closely matches temperature scaling for (single-model) calibration [21].

This temperature tuning automatically guards against defective Duos. If $f_{\text{small}}$ is too inaccurate, then the Duo can effectively revert to $f_{\text{large}}$ through the weighting $T_{\text{large}} = 1$, $T_{\text{small}} = 0$. Conversely, any non-trivial weighting (i.e. $T_{\text{small}} > 0$) implies that the Asymmetric Duo provides benefit over $f_{\text{large}}$ since it improves negative log likelihood on the validation set.

**Ablations.** To understand how our proposed asymmetric Duo affects performance on downstream tasks, our experiments and analyses consider the following ablations:

1. **Unweighted Duos** equally weight the predictions of both models without regard for their asymmetry, similar to deep ensembles. It is equivalent to setting $T_{\text{large}} = T_{\text{small}} = 0.5$: This ablation assesses the importance of a weighted average for aggregating logits.

2. **UQ Only Duos** only use Duo predictions for the uncertainty estimate, using the base model's class prediction:

$$\hat{Y}_{\text{Duo}} = \hat{Y}_{\text{large}} = \arg\max_i [f_{\text{large}}(X)]_i, \qquad \text{unc}(f_{\text{Duo}}(X)) = 1 - [\sigma(f_{\text{Duo}}(X)]_{\hat{Y}_{\text{large}}(X)}.$$

This ablation isolates the effects of Asymmetric Duos on correctness prediction and selective classification tasks, demonstrating whether any benefits to performance can be attributed to better uncertainty quantification versus more accurate class predictions.

## 3.2 Measuring the Computational Cost of Asymmetric Duos

The cost of a Duo versus a single model is dictated by the degree of asymmetry in size. How to measure size is non-trivial, because no one property fully captures a deep network's size or complexity. Parameter count, network depth, floating point operations per forward pass (FLOPs), and throughput each reflect different aspects of model capacity and computational cost [25, 52, 56]. To focus on practical use we measure computation. We profile FLOPs due to their consistency across hardware in Section 4, and we profile throughput as measure of real-world time efficiency in Appendix B. Asymmetric duos deliver efficient improvement for either measure of computational cost.

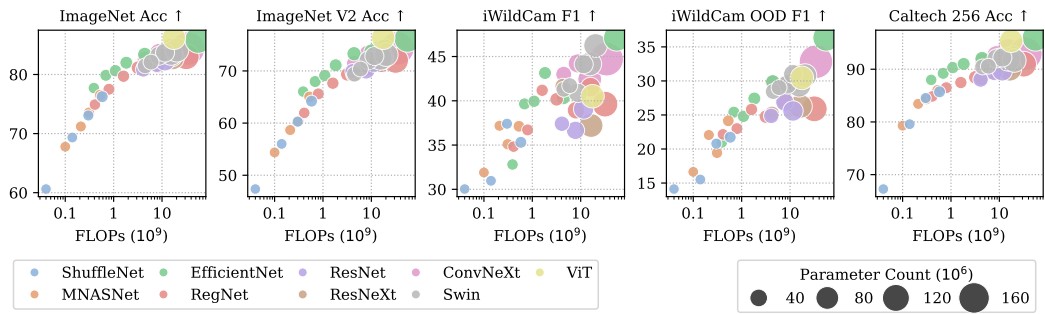

Figure 2: **Model Size Correlates with Accuracy.** Performance improves with higher FLOPs and parameter counts across many models on multiple benchmarks. Our Duos allow us to adjust size: combining a bigger $f_{\text{large}}$ with a littler $f_{\text{small}}$ adds just a little computation, or more, as we choose.

We measure asymmetry as the relative increase in FLOPs over using $f_{\text{large}}$ in isolation:

$$\text{FLOPs Balance} = \% \text{ Computation Increase} = \text{FLOPs}(f_{\text{small}})/\text{FLOPs}(f_{\text{large}}).$$

## 4 Experiment Setup

We evaluate Asymmetric Duos in the context of image classification, measuring their effects on accuracy, uncertainty quantification, and downstream decision-making. To demonstrate their "out-of-the-box" nature, we focus on Duos that combine models from popular pre-trained models available from the `torchvision` [37, 43] and `timm` [59] "model zoos."

**Datasets.** We benchmark **ImageNet** [8] for large-scale evaluation, and **Caltech 256** [20] and **iWildCam** [3] for measuring transfer to new domains. As accuracy and uncertainty quantification tend to degrade under distribution shifts [42], we further evaluate with **ImageNet V2** [48] and **iWildCam-OOD** [27] to test robustness under such shifts.

**Models.** For $f_{\text{large}}$ architecture and parameters, we consider pre-trained models that 1. consistently achieve high performance on fine-tuning benchmarks [e.g. 18] and 2. are commonly used in practice for various computer vision tasks. We therefore experiment with four strong $f_{\text{large}}$ that cover convolution and attention architectures: a **ConvNeXt-Base** [35] pre-trained on ImageNet, a **SwinV2-Base** [34] pre-trained on ImageNet, an **EfficientNet-V2-L** [53] pre-trained on the larger-scale ImageNet21K, and a **CLIP ViT-B** [10, 45] pre-trained on the larger still LAION-2B [50]. These span standard scale (ImageNet) to extremely large scale (LAION). For $f_{\text{small}}$ models, we consider smaller architectures from `torchvision` and `timm` [24, 33, 46, 52, 54, 61, 62] (see Appendix A for a complete list). Within a given model family (e.g. ResNet), we choose models of varying size (e.g. ResNet-34, ResNet-50, ResNet-101, etc.) to produce Duos with various levels of asymmetry.

**Fine-Tuning and Evaluation.** For our downstream datasets evaluations, we fine-tune our models following the LP-FT recipe [28] with standard unweighted Cross-Entropy Loss, AdamW [36] optimizer, linear warm-up with cosine annealing learning rate scheduler, and perform hyperparameter tuning for all models over learning rate and weight decay on a withheld validation set.

We calibrate all fine-tuned models by temperature scaling [21], minimizing the negative log likelihood on the validation set using L-BFGS, to improve baseline uncertainty quantification.

**The Size-Accuracy Relationship of Large and Small Models.** Figure 2 summarizes the class prediction accuracy of all the pre-trained and fine-tuned models that we use to construct Asymmetric Duos. We report accuracy for ImageNet, ImageNet V2, and Caltech 256 and Macro F1 Score for iWildCam2020-WILDS to account for the significant class imbalances present [27].

Consistent with prior work [18, 38], we find a strong correlation between model size (as measured by FLOPs/parameter counts) and in-distribution (InD) and out-of-distribution (OOD) accuracy. In

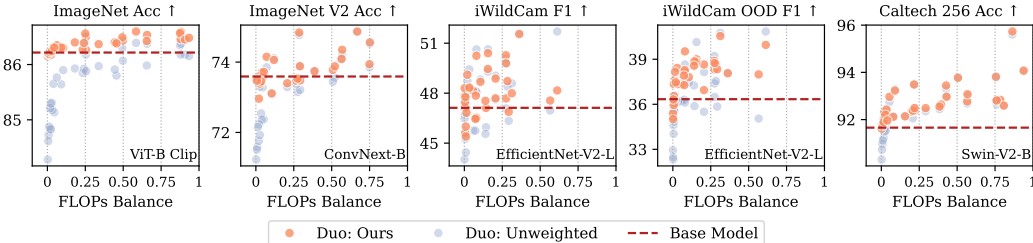

Figure 3: **Class Prediction (Accuracy and F1 ↑)** as a function of FLOPs balance. (Balance = 0 corresponds to cost of a single $f_{\text{large}}$ model; 1 corresponds to the cost of a $m = 2$ ensemble.) Asymmetric Duos almost always increase accuracy over a single $f_{\text{large}}$, even for Duos that combine $f_{\text{large}}$ with a $f_{\text{small}}$ that is $1/10^{\text{th}}$ the size. The learned temperature weighting is crucial to achieve this performance; if the predictions of $f_{\text{large}}$ and $f_{\text{small}}$ are averaged with equally weight (Duo: Unweighted) then imbalanced Duos may have significantly worse performance than single models.

the next section, we investigate to what degree combining pairs of models into Asymmetric Duos allow us to practically traverse the size/performance tradeoff.

## 5  Experiment Results

Here we present the performance of Asymmetric Duos, as measured by accuracy, uncertainty quantification, and selective classification performance, as a function of computation (i.e. Duo FLOPs balance). For each benchmark, we form Asymmetric Duos using all possible combinations of $f_{\text{large}}$ and $f_{\text{small}}$ architectures outlined in Section 4. For visual clarity, this section only contains results for a single $f_{\text{large}}$ on each dataset. (We choose different $f_{\text{large}}$ architectures for each dataset for diversity of results within this section.) We note that the results in this section are representative of all $f_{\text{large}}$ architectures (see Appendix G for the full set of results).

### 5.1  Duos as Classifiers – Accuracy & Macro F1 Score

To determine if the addition of $f_{\text{small}}$ can enhance class prediction, we compare the accuracy of Asymmetric Duos to that of a single $f_{\text{large}}$ model in isolation. We report top-1 accuracy (ImageNet V2 and Caltech 256) and Macro F1 score (iWildCam) in Figure 3 plotted against FLOPs balance.

Overall, we find that Asymmetric Duos can yield significant improvements in accuracy/F1 over $f_{\text{large}}$. We note that the smallest Duos do not provide an accuracy advantage, and occasionally produce minor performance drops. This result is expected, and arguably the lack of major performance degradations is surprising given the significantly lower accuracy of the smallest $f_{\text{small}}$ models. However, after the FLOPs balance reaches $10\%$, which in many settings may be negligible additional computation, Duos provide consistent accuracy improvements across all datasets.

We find that the proposed temperature weighting scheme is necessary for these improvements in accuracy. In Appendix D we report the learned temperatures of all Duos, finding larger $f_{\text{large}}$ weights when $f_{\text{small}}$ is smaller, with more even weights as balance increases. Our ablation further confirms this hypothesis. We observe that Duos that equally average the $f_{\text{large}}$ and $f_{\text{small}}$ predictions (Duo: Unweighted) significantly underperform $f_{\text{large}}$ when the balance is near zero.

In Appendix G we also report results for probabilistic measures of accuracy; namely Brier Score and negative log likelihood. The results for these metrics follow similar trends.

### 5.2  Predictive Uncertainty Separability - Correctness Prediction AUROC

Predictive uncertainty estimates are often judged by their calibration, as measured by Expected Calibration Error (ECE) [e.g. 21, 42]. Because we apply temperature scaling throughout our experiments, the $f_{\text{large}}$ models are largely calibrated to begin with, and so Asymmetric Duos provide little improvement. (See Appendix G for results.) We argue that this is not a failure of Duos but rather a success of temperature scaling.

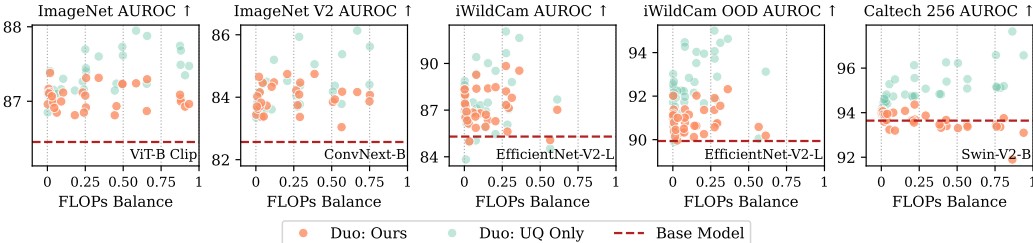

Figure 4: **Correctness prediction as measured by AUROC** (↑), which captures the separability of correct and incorrect predictions by uncertainty. In almost all cases, Asymmetric Duos achieve higher AUROC than corresponding $f_{\text{large}}$ models in isolation, even when using a negligible amount of additional computation. We confirm that this increase cannot be attributed to accuracy improvements alone: our ablation (Duo: UQ only) that uses the Duo's uncertainty measure to separate the $f_{\text{large}}$ class prediction produces a comparable AUROC increase.

We instead choose to measure the quality of Duo uncertainty estimates based on their correlation with the correctness of class predictions. Intuitively, a good uncertainty estimate should be able to separate correct versus incorrect class predictions. To that end, we report correctness prediction Area Under the Receiver Operating Characteristic curve (AUROC) for Duos and for single $f_{\text{large}}$ as a function of FLOPs balance. Randomly assigned uncertainty measure would yield an AUROC score of $0.5$, while a perfectly separable uncertainty measure would yield $1.0$.

In Figure 4 we find that Asymmetric Duos consistently yield higher AUROC than $f_{\text{large}}$. Surprisingly, we observe performance improvements even when the FLOPs balance is near $0$. This finding suggests that AUROC can be increased without much additional computation. Performance does not correlate with additional computation (i.e. higher FLOPs balance), but all Duos provide benefit.

Our ablation demonstrates that these AUROC improvements cannot solely be attributed to the increased accuracy afforded by Duos. Recall that the UQ Only Duo variant uses the Duo's uncertainty estimate while retaining the class prediction of $f_{\text{large}}$. Any AUROC benefits must therefore be attributed to improvements in the uncertainty, as accuracy remains unchanged. We observe that UQ Only Duos obtain significantly better AUROC score compared to their base models, showing that the sidekick model is effective at re-ranking uncertainties for the class prediction of $f_{\text{large}}$, confirming that Duos provide meaningful improvements in uncertainty separability in addition to accuracy.

### 5.3 Duos as selective classifiers – AURC

To study the effect of Duo uncertainty estimates on downstream decision-making tasks, we evaluate Asymmetric Duos in the context of selective classification. Selective classification allows a model to abstain from prediction when the uncertainty is high, deferring uncertain cases to experts in effective human-in-the-loop systems [16]. The proportion of the test set with uncertainty below a given threshold is referred to as coverage, and performance on this subset is measured using the domain-specific loss, which in our case is the 0-1 loss analogous to classification error rate. As the uncertainty threshold decreases, the model becomes more cautious, abstaining from more inputs and reducing coverage while aiming to improve accuracy on this smaller covered subset. We emphasize that high-quality uncertainty estimates are necessary for selective classifiers to work in practice.

We evaluate selective classification performance using the Area Under the Risk Coverage (AURC) metric [17]. AURC accumulates error rates at all coverage levels to measure a model's overall potential as a selective classifier. As with correctness prediction, the AURC metric conflates accuracy and uncertainty quantification, as different predictors achieve different accuracy at full coverage (entire test set accuracy). To that end, we evaluate our UQ Only Duo ablation in addition to the proposed approach to bypass this potential confound.

In Figure 5, we observe that Duos provide substantial boost to AURC at all computation levels, even when the FLOPs balance is fractional. Analogous to our correctness prediction results, UQ Only Duos also achieve improved AURC scores at low FLOPs balance, demonstrating that these improvements cannot be attributed to accuracy alone. This is a direct result of our findings on

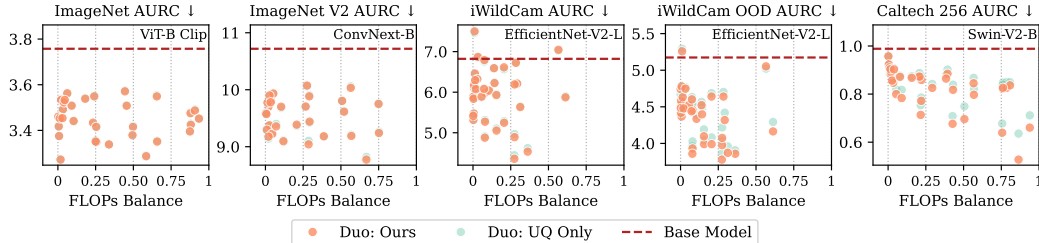

Figure 5: **Selective classification performance as measured by AURC (↓)**, which averages the error across classification coverage levels/abstaining rates. Duos significantly improve this metric while adding as little as 10% additional computation. As with our correctness prediction results (Figure 4), our UQ Only ablation confirms that these improvements cannot be solely attributed to increases in accuracy.

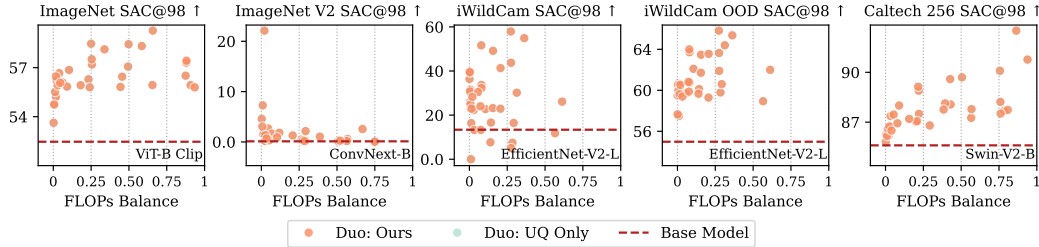

Figure 6: **Selective classification performance as measured by SAC (↑)**, which measures the classification coverage under a fixed accuracy constraint. Duos achieve significantly higher coverages at all levels of computation. As with our correctness prediction results (Figure 4), our UQ Only ablation confirms that these improvements cannot be solely attributed to increases in accuracy.

correctness prediction with AUROC. Together, these results also confirm that smaller models can effectively re-rank the uncertainties of larger models. In contrast, many prior methods for selective classification sacrifice non-selective predictive power [14], resulting in unfavorable trade-offs in practical metrics such as AURC.

### 5.4 Duos as selective classifiers – Selective Accuracy Constraint

To further evaluate Duos in the selective classification context, we report the Selective Accuracy Constraint (SAC) metric for all Duos and $f_{\text{large}}$ models. SAC is a practical evaluation metric for risk-sensitive applications with a guarantee. It measures the maximum coverage a predictor can achieve while maintaining a fixed target accuracy, making it well-suited for settings where reliability is critical. Prior work in selective classification has shown that leveraging softmax response uncertainty allows deep networks to achieve high accuracy on a subset of confidently predicted data by abstaining on uncertain cases [14, 16]. Here we show duos boost coverage at the ambitious accuracy levels of 98% across all of our benchmarked datasets.

In Fig 6, we observe that our proposed Asymmetric Duos achieve higher coverage at minimal FLOPs balances. This finding complements our previous results with the AURC metric. Furthermore, the UQ Only ablation almost always match our proposed method, further implying that the improvements in coverage at high accuracy levels stems from re-ranked uncertainties rather than accuracy increases. Altogether, these results suggest that Asymmetric Duos are a practical choice in high-risk decision-making applications.

### 5.5 Soup with a (Side)kick: Combining Model Soups and Duos

We show the Asymmetric Duo framework is agnostic to model type, compatible with other methods, and compounds uncertainty quantification improvements by making a duo with a model soup [60]. For this experiment, we fine-tuned sixteen ConvNeXt Base models on Caltech 256 starting from

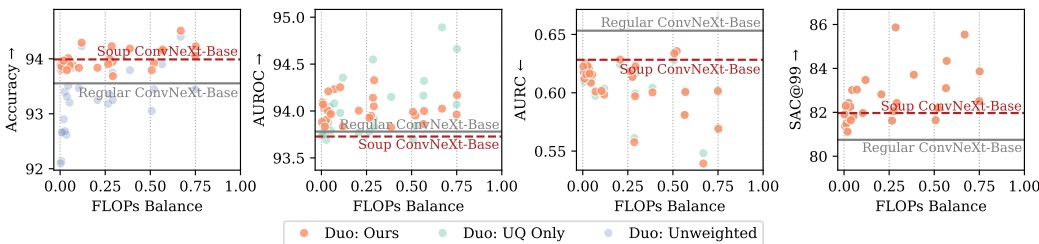

Figure 7: **Asymmetric Duos are compatible with Soups.** A model soup (red line) achieves better performance than a standard single model (grey line) across accuracy, AURC, and SAC@98. Our Duos provide further improvement by incorporating the soup as $f_{\text{large}}$.

the same torchvision pre-trained model using different training hyperparameters. We then mixed a greedy model soup for use as $f_{\text{large}}$ in our Asymmetric Duo. In Figure 7, we observe that Duos are compatible with soups and other methods of making better base and sidekick models.

### 5.6 Comparisons between Asymmetric Duos and Deep Ensembles

We show that Asymmetric Duos (AD) can match the performance of classic deep ensembles (DE) with a significantly lower compute budget. We construct ensembles of ResNet50s (RN50) trained on the ImageNet dataset following the "pool-then-calibrate" strategy from [47]. We compare against ADs that pair a RN50 ($f_{\text{large}}$) with MNASNet0.75 (MN.75), ShuffleNet V2 2.0× (SN2.0x), and EfficientNet-B2 (ENB2) models, yielding FLOPs Balances (FB) of 0.05, 0.14, and 0.27, respectively.

Table 1: **Base-model-controlled comparison between Asymmetric Duos and deep ensembles.**

| Method | Balance↓ | Acc↑ | Brier↓ | NLL↓ | ECE↓ | AUROC↑ | AURC↓ |
|---|---|---|---|---|---|---|---|
| RN50 (Temp Scaled) | 0 | 79.82±0.51 | 2.90±0.05 | 84.06±3.38 | 3.39±1.39 | 86.62±0.49 | 5.72±0.38 |
| [AD] RN50+MN.75 | 5 | 79.88±0.42 | 2.87±0.05 | 80.45±2.17 | 3.25±0.61 | 87.11±0.29 | 5.42±0.19 |
| [AD] RN50+SN2.0x | 14 | 80.32±0.32 | 2.81±0.04 | 78.93±1.78 | **2.62±0.76** | 87.00±0.27 | 5.34±0.17 |
| [AD] RN50+ENB2 | 27 | **81.87±0.19** | **2.60±0.03** | **71.47±0.84** | 2.80±0.48 | 87.15±0.26 | 4.80±0.17 |
| [DE] RN50 (m=2) | 100 | 81.12±0.29 | 2.72±0.04 | 77.12±1.90 | 3.54±1.33 | 87.19±0.30 | 4.99±0.24 |
| [DE] RN50 (m=5) | 400 | **81.97±0.13** | **2.61±0.01** | 73.16±0.90 | 3.79±0.76 | **87.62±0.11** | **4.48±0.09** |

Table 1 shows that Asymmetric Duos quickly approach performance of $m = 5$ Deep Ensembles (e.g., $m = 5$ RN50s) across most metrics, despite using $4\times$ fewer FLOPs. We attribute this to architectural diversity from heterogeneity [19] and Duos' simple but effective temperature-based aggregation. If we allow Asymmetric Duos to match the computational budget of $m = 5$ Deep Ensembles, then Duos become even more advantageous. See Appendix E for a FLOPs-controlled comparison.

## 6 Discussion

In this work, we proposed the Asymmetric Duos framework, which yield consistent improvements across a range of predictive performance, uncertainty quantification, and decision making metrics. While these improvements may also be achievable through deep ensembles, Asymmetric Duos offer a substantially more efficient alternative, requiring only a fraction of the additional computation cost of their larger member model. Moreover, the framework is inherently compatible with fine-tuning workflows, making it a practical and scalable strategy in modern deployment scenarios.

**Methods related to Duos in construction.** Gontijo-Lopes et al. [19] used a similar construction of heterogeneous $m = 2$ deep ensembles to study the effect of predictive diversity induced by models pre-trained on different datasets. Our work differs mainly by aggregating deep network pairs of vastly different sizes as opposed to different architectures with similar accuracy, diving deeper into a smaller member model's effects on a Duo's uncertainty, and of course our fine-tuning the pairs on the same data.

Rahaman and Thiery [47] studied the combination of temperature scaling [21] and deep ensembles [29] and found that deep ensembles constructed by pool-then-calibrate or pool-with-calibrate strategies outperform naive deep ensembles and those following a calibrate-then-pool strategy on proper scoring rules. Our proposed Asymmetric Duos follow a similar construction order to a pool-with-calibration strategy, but fundamentally differ in motivation. As shown with our Unweighted Duos ablation, the asymmetry between $f_{\text{large}}$ and $f_{\text{small}}$ makes a pool-with-calibrate strategy necessary in preventing degradation in accuracy. Whereas Rahaman and Thiery [47] focuses on such constructions' effects on calibration, we put a unique emphasis on uncertainty quantification metrics like correctness prediction and selective classification.

**Limitations and future directions.** Our experiments are limited to the image classification task. While our findings show strong performance and computational efficiency on a total of five test sets, it remains unexplored how Asymmetric Duos perform in other tasks and modalities, such as image segmentation, regression, and natural language processing. Exploring these extensions could reveal broader applicability of the framework.

This work serves as an initial step toward understanding the aggregation strategies of deep networks of different sizes and capacities. While we focused on a small set of interpretable aggregation strategies for both point prediction and uncertainty quantification, there remains a large space of unexplored methods. Notably, our finding that simple validation-based weighting can yield effective ensembles from models with vastly different accuracies is itself a surprising and underexplored result. Future work may benefit from exploring other learning-based, adaptive weighting schemes for member aggregation, and may even call them Dynamic Duos.

## Acknowledgements

GP and ES are supported by Canada CIFAR AI Chairs. We acknowledge the support of the Natural Sciences and Engineering Research Council of Canada (NSERC: RGPIN-2024-06405). Resources used in preparing this research were provided, in part, by the Province of Ontario, the Government of Canada through CIFAR, and companies sponsoring the Vector Institute.

We thank Mathias Lécuyer for his helpful review and feedback on the experiments and exposition.

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

# A Experimental Details

## A.1 Model Architectures

We select popular, recent, and performative pre-trained models from `torchvision` [37] and `timm` [59] to construct our Duos, to make sure Duos are compatible with various recent architectures. From `torchvision`, we experiment with EfficientNet [52], EfficientNet V2 [53], MNASNet [54], ConvNeXt [35], ResNet [24], ResNeXt [61], RegNet [46], ShuffleNet-V2, Swin [62], and Swin (V1 & V2) [33, 34]. From `timm`, we get an Efficientnet-V2-L pre-trained on ImageNet 21k [8], and a ViT-B-16 [10] pre-trained on LAION 2B [50].

## A.2 Training Procedure

For all our fine-tuned experiments we follow the LP–FT recipe from Kumar et al. [28]: we first train only the final classification head (also known as Linear Probing, or LP) for a few epochs (8 for both Caltech 256 and iWildCam) using cross-entropy loss and AdamW, then unfreeze the entire network and Fine-Tune (FT) for more epochs (16 on Caltech 256, 12 on iWildCam) under the same loss and optimizer.

Our FT procedure employs a sequential schedule (linear warm-up followed by cosine annealing), and we perform hyper-parameter search over learning rate in $[1 \times 10^{-6}, 3 \times 10^{-4}]$ and weight decay in $[1 \times 10^{-8}, 1 \times 10^{-5}]$, picking the best trial by validation score. We use a batch size of 128 for Caltech 256 and 16 for iWildCam. All models are trained on NVIDIA L40S GPUs. Random Augmentation [5] is used to augment training samples during the FT phase. The best LP checkpoint initializes FT, and the top FT model by validation performance is carried forward into our Duo evaluations.

## A.3 Datasets

Here we provide descriptions of each dataset used and how train-val-test splits are chosen in our experiment.

**ImageNet** [8] contains 1,000 classes and over a million training samples. All the pretrained models used in our experiment have been trained on ImageNet as a last step. Since ImageNet never released the official test set, researchers split the official validation set into val and test splits. Since we don't need to fine-tune models on ImageNet for our experiment, our only usage of a validation set is to temperature scale single models and to tune temperature weightings for Asymmetric Duos. We use only 5% of the official test split as our validation set to show how data-efficient the temperature-weighting step is.

**ImageNet V2** [48] is a careful reproduction of the original ImageNet, but nevertheless showed model performance degradation, indicating distribution shifts. Since this dataset is used solely for evaluation, the entire dataset is treated as an OOD test set for ImageNet models and Duos.

**iWildCam (OOD)** [3, 27] is a camera-trap dataset with 203,029 training images containing 182 classes of animals. The IND and OOD are determined by distinctive camera traps, since different locations vary largely in vegetation and backgrounds, resulting in poor generalization to new camera traps. We use the official IND val split for iWildCam, which contains 7315 images captured by the same set of camera traps as the training set.

**Caltech 256** [20] contains 30,607 images spanning 256 object classes and a clutter class. We use a random 15% split of the dataset as val and another 15% as test. The same validation set was used for both hyper-tuning and temperature tuning.

## A.4 Greedy Soup Setup

For our Greedy Soup experiments, we independently fine-tune 16 ConvNeXt-Base models initialized from the same backbone with different hyperparameters over the same search space as our regular tuning procedure on Caltech 256 starting from the same pre-trained weights from `torchvision`, and greedily average their weights to improve performance on the validation set following the procedure outlined by Wortsman et al. [60]. The greedy soup in Figure 7 averages three of the sixteen trials.

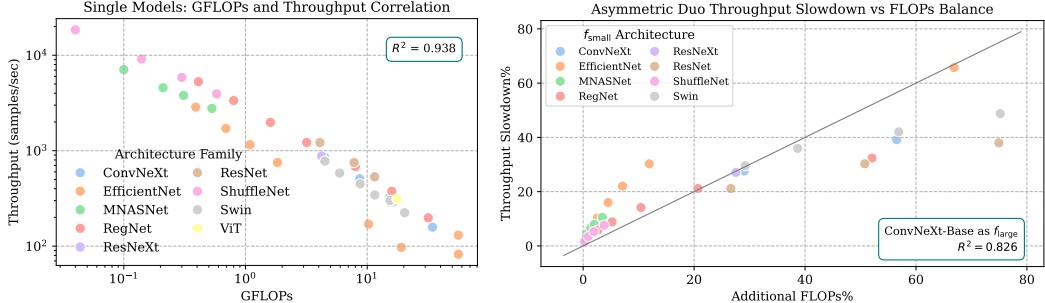

Figure 8: Comparison of throughput versus FLOPs as measures of computational cost/model size for single models (left) and ConvNeXt-Large-based Duos (right). In both cases we find large correlation between the two measures.

### A.5 Deep Ensemble Setup

All $m = x$ DEs in Section 5.6 and Appendix E are constructed by randomly selecting 32 subsets of size $x$ from 10 independently trained ResNet-50 checkpoints from timm and torchvision, following the "pool-then-calibrate" strategy of Rahaman and Thiery [47]. Because our temperature-based aggregation strategy for Duos naturally incorporates calibration, this setup enables a fair comparison on NLL, Brier, and ECE metrics.

All ADs used for comparison in Section 5.6 are formed by pairing one ResNet-50 checkpoint as $f_{\text{large}}$ with a sidekick $f_{\text{small}}$ drawn from torchvision.

## B    Alternative Measure of Computation (Throughput)

As discussed in Section 3.2, deep networks have several reasonable characterizations of size/computational cost, including parameter counts, FLOPs, depth, and throughput. We explore FLOPs in the main text due to its consistency across hardware and the intuitive interpretation of the FLOPs balance as the additional computation cost as a percentage of the larger member model's computation cost. However, FLOPs fail to consider architectures' parallelization differences, which play a factor in a model's speed during deployment. Here we show how our FLOPs balance notion would translate into throughput, on a single NVIDIA V100 Volta GPU. Figure 8 shows a clear linear correlation between throughput and FLOPs, with small variations among architecture families. For single models, the coefficient of determination for both metrics is $0.938$; for Duos it is $0.795$. This result implies that the fractional additional FLOPs repeatedly emphasized in our main text translate directly to wall-clock-time efficiency.

## C    Alternative Measures of Uncertainty–Entropy

Throughout the paper we use softmax response to quantify the predictive uncertainty of single models and Duos. (Eq. 1). Here we demonstrate that our results are robust to this choice. We replicate our main results from Section 5 where we instead measure uncertainty via the entropy of the predictive distribution, another common measure used in practice. Figure 9 shows AUROC, AURC, and SAC@98 performance under entropy-based uncertainty quantification. We observe similar performance trends to our softmax response-based results in Figures 4, 5, and 6.

## D    Analysis of Temperature Weightings

In Figure 10 we plot the ratio of the temperatures $T_{\text{large}}/T_{\text{small}}$ for Asymmetric Duos of various FLOPs balances for all $f_{\text{large}}$ models on Caltech 256 to study how temperatures are assigned for fine-tuned models. A ratio $\gg 1$ implies that $f_{\text{small}}$ does not contribute to the Duo prediction and that the Duo functions like a single model. A ratio that is closer to 1 implies that $f_{\text{small}}$ and $f_{\text{large}}$ both meaningfully contribute to the Duo and that we should expect higher performance than from

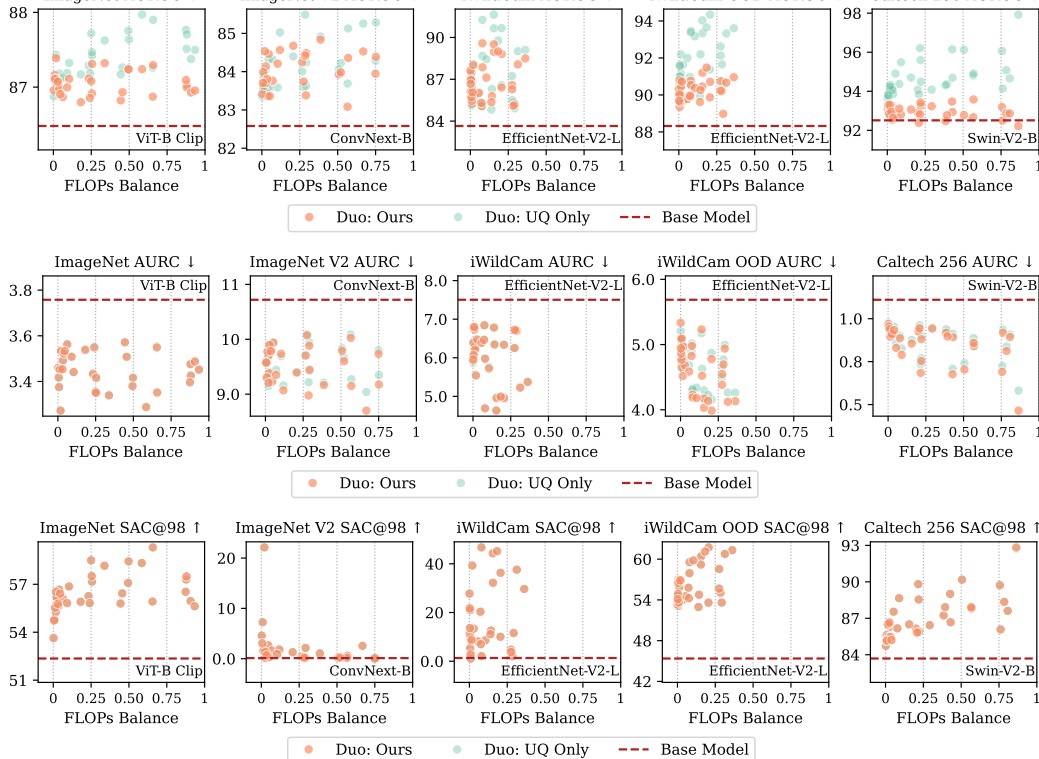

Figure 9: Using entropy to measure uncertainty yields similar results on Asymmetric Duos' correctness prediction and selective classification metrics.

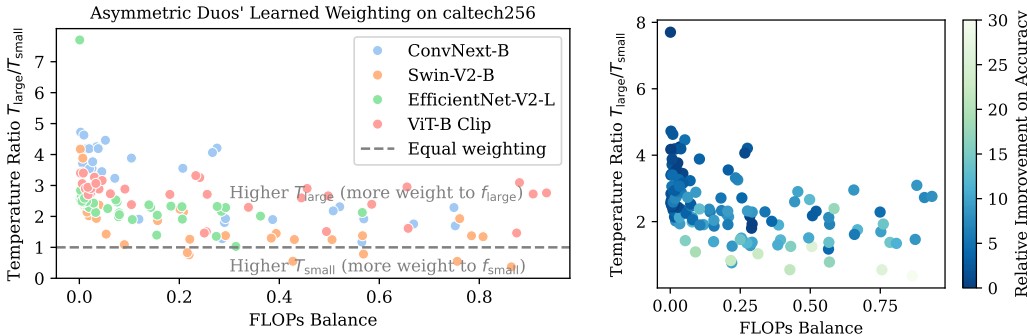

Figure 10: (Left) Ratio of learned temperatures $T_{\text{large}}/T_{\text{small}}$ for Asymmetric Duos on the Caltech dataset. With small FLOPs balances, $f_{\text{large}}$ is weighted up to $7\times$ more than $f_{\text{small}}$. As the FLOPs balance increases, the weighting becomes more even. (Right) When FLOPs balances is close to 0, Asymmetric Duos significantly up-weight $f_{\text{large}}$ to prevent performance degradation; as FLOPs balances increase, Asymmetric Duos learn to assign more weights to $f_{\text{small}}$ for higher improvements.

what $f_{\text{large}}$ achieves in isolation. As expected, the temperature for $f_{\text{large}}$ is larger than that of $f_{\text{small}}$ for small FLOPs balances. As the balance grows, the Duo members are weighted more equally.

# E  Asymmetric Duos versus Deep Ensembles

Section 5.6 compares Asymmetric Duos and pool-then-calibrate DEs with the same base model, finding that Duos nearly match the performance of ensembles with a fraction of compute. In this section, we compare ADs and DEs under a fixed computational budget, which allows the Duo to use a much larger $f_{\text{large}}$ than the DE base model.

Table 2: **FLOPs-controlled comparison between Asymmetric Duos and deep ensembles.** AD[$m$,$m+1$] denotes Duos with total compute falling between $m$ and $m+1$ times the FLOPs of a base ResNet-50 (RN50) model. RN50 AD[1,2] includes only Duos with RN50 base models for base-controlled comparison against DEs.

| Method | FLOPs(G) | Acc↑ | Brier↓ | NLL↓ | ECE↓ | AUROC↑ | AURC↓ |
|---|---|---|---|---|---|---|---|
| RN50 (temp-scaled) | 4.1 | 79.82±0.51 | 2.90±0.05 | 84.06±3.38 | 3.39±1.39 | 86.62±0.49 | 5.72±0.38 |
| RN50 AD[1,2] | [4.1,8.2) | **81.5±0.91** | **2.7±0.12** | **73.0±3.87** | **2.6±0.95** | 86.8±0.28 | **5.0±0.32** |
| RN50 DE(m=2) | 8.2 | 81.08±0.29 | 2.72±0.04 | 77.19±1.66 | 3.40±1.32 | **87.18±0.31** | 5.01±0.24 |
| AD[4,5] | [16.4,20.5) | **84.8±1.2** | **2.2±0.1** | **59.4±5.4** | **2.9±0.7** | 86.9±0.7 | **3.9±0.4** |
| RN50 DE(m=5) | 20.5 | 81.95±0.15 | 2.61±0.01 | 72.77±0.84 | 3.94±0.73 | 87.56±0.13 | 4.53±0.13 |
| AD[7,8] | [28.7,32.8) | **86.3±0.2** | **2.0±0.0** | **53.0±0.9** | 3.0±0.4 | **87.3±0.3** | **3.4±0.1** |
| RN50 DE(m=8) | 32.8 | 82.19±0.06 | 2.57±0.01 | 71.78±0.41 | **2.93±0.17** | 87.64±0.04 | 4.39±0.02 |

Table 2 shows that ADs (e.g., AD$[4, 5]$) considerably outperform DEs with similar test-time FLOPs cost (e.g., RN50 DE with $m = 5$) on most metrics. Notably, while DE performance tends to saturate as $m$ increases, ADs continue to improve thanks to the stronger base model at higher FLOPs budgets. This suggests a simple and effective strategy: when deploying Asymmetric Duos, choose the most performant $f_{\text{large}}$ that your budget allows to maximize both accuracy and uncertainty quantification metrics. The setups on DEs are identical to Table 2 and is described Appendix A

# F  Evaluating Asymmetric Duos against scalable Bayesian Methods

Bayesian approaches have been prominent for uncertainty quantification tasks, at times performing competitively against deep ensembles. However, similar to deep ensembles, many Bayesian methods still struggle in scalability, making them hard to compare with methods like Asymmetric Duos. Here we evaluate Asymmetric Duos with two practical Bayesian methods, Improved Variational Online Newton (IVON) [51] and Laplace Approximation [6].

## F.1  Improved Variational Online Newton (IVON)

The recently proposed IVON [51] was the first variational Bayesian objective optimizer that matched AdamW [36] in Accuracy when applied to large-scale training, in both from-scratch and fine-tuning settings. We benchmark our Asymmetric Duos against a ResNet50 fine-tuned with IVON optimizer on Caltech256.

Table 3: **Base-model-controlled comparison between Asymmetric Duos and IVON.**

| Method | Acc↑ | Brier↓ | NLL↓ | ECE↓ | AUROC↑ | AURC↓ |
|---|---|---|---|---|---|---|
| Baseline ResNet50 | 87.72 | 7.60 | 67.90 | 7.56 | 91.81 | 1.92 |
| Temp-Scaled ResNet50 | 87.72 | 6.88 | 49.54 | 1.22 | 91.44 | 2.01 |
| IVON (deterministic) | 87.98 | 6.63 | 45.33 | 1.89 | 91.92 | 1.83 |
| IVON (2 samples) | 87.23±0.33 | 7.21±0.09 | 49.44±0.58 | 1.59±0.26 | 91.33±0.27 | 2.09±0.06 |
| IVON (4 samples) | 87.98±0.17 | 6.91±0.05 | 47.08±0.37 | 2.60±0.22 | 91.40±0.27 | 1.89±0.05 |
| IVON (8 samples) | 88.19±0.16 | 6.77±0.02 | 45.88±0.22 | 3.10±0.20 | 91.65±0.28 | 1.81±0.02 |
| IVON (16 samples) | 88.41±0.17 | 6.72±0.01 | 45.52±0.14 | 3.36±0.19 | 91.47±0.29 | 1.78±0.01 |
| IVON (32 samples) | 88.43±0.13 | 6.68±0.02 | 45.17±0.15 | 3.46±0.15 | 91.58±0.10 | 1.77±0.02 |
| **Asymmetric Duos [1,2]** | **89.75±1.17** | **5.80±0.60** | **40.73±4.56** | **0.97±0.28** | **91.58±0.53** | **1.52±0.18** |

Table 3 shows that IVON works well out of the box, outperforming standard AdamW with minimal training overhead. Its sampling-based inference further improved over the deterministic version,

though like Deep Ensembles, it introduces an test-time cost with diminishing returns. In contrast, Asymmetric Duos built on ResNet-50 achieve better performance across all metrics with only a fractional increase in compute. Furthermore, similar to the compatibility shown between soup and Asymmetric Duos in Section 5.5, IVON, as a great plug-and-use uncertainty-aware optimizer, would fit nicely with our Asymmetric Duos framework.

## F.2   Laplace Approximation

Laplace Approximation(LA) is another practical and competitive Bayesian method, approximating the posterior with a Gaussian distribution centered at any regularly trained model [6]. We benchmark our Asymmetric Duos against Laplace Approximation on a Swin-V2-S trained on Caltech256, as models with larger final feature dimensions are computationally infeasible for LA given our hardware limits. For the Laplace Approximation method, we follow the original authors and use a kron hessian structure to approximate the posterior of the last-layer of the model. We limit FLOPs Balance (FB) to be $0.1 < \text{FB} < 0.3$ for Duos to emphasize its computational efficiency.

Table 4: **Base-model-controlled comparison between Asymmetric Duos and Laplace Approximation (LA).**

| Method | Acc↑ | F1↑ | Brier↓ | NLL↓ | ECE↓ | AUROC↑ | AURC↓ | SAC@98↑ |
|---|---|---|---|---|---|---|---|---|
| Swin-V2-S baseline | 91.7 | 90.5 | 5.3 | 49.1 | 5.5 | **93.8** | 1.0 | 85.3 |
| Swin-V2-S temp-scaled | 91.7 | 90.5 | 4.7 | 33.3 | 1.5 | 93.6 | 1.0 | 85.6 |
| Swin-V2-S LA | 92.2 | 91.4 | 4.9 | 41.6 | 4.9 | 93.4 | 0.9 | 86.5 |
| Asymmetric Duos | **93.0±0.3** | **92.2±0.3** | **4.0±0.2** | **26.9±1.3** | **1.1±0.2** | 93.4±0.2 | **0.8±0.0** | **87.9±0.7** |

Table 4 shows that although Laplace Approximation slightly improve upon baseline on metrics such as AUROC and AURC, it falls short in absolute performance gains compared to Asymmetric Duos across most metrics.

# G Complete Experimental Results

Here we present full results for all 4 $f_{\text{large}}$ models across their benchmarked datasets.

**CLIP ViT-B Pretrained on ImageNet 21K** is benchmarked as $f_{\text{large}}$ on all five datasets.

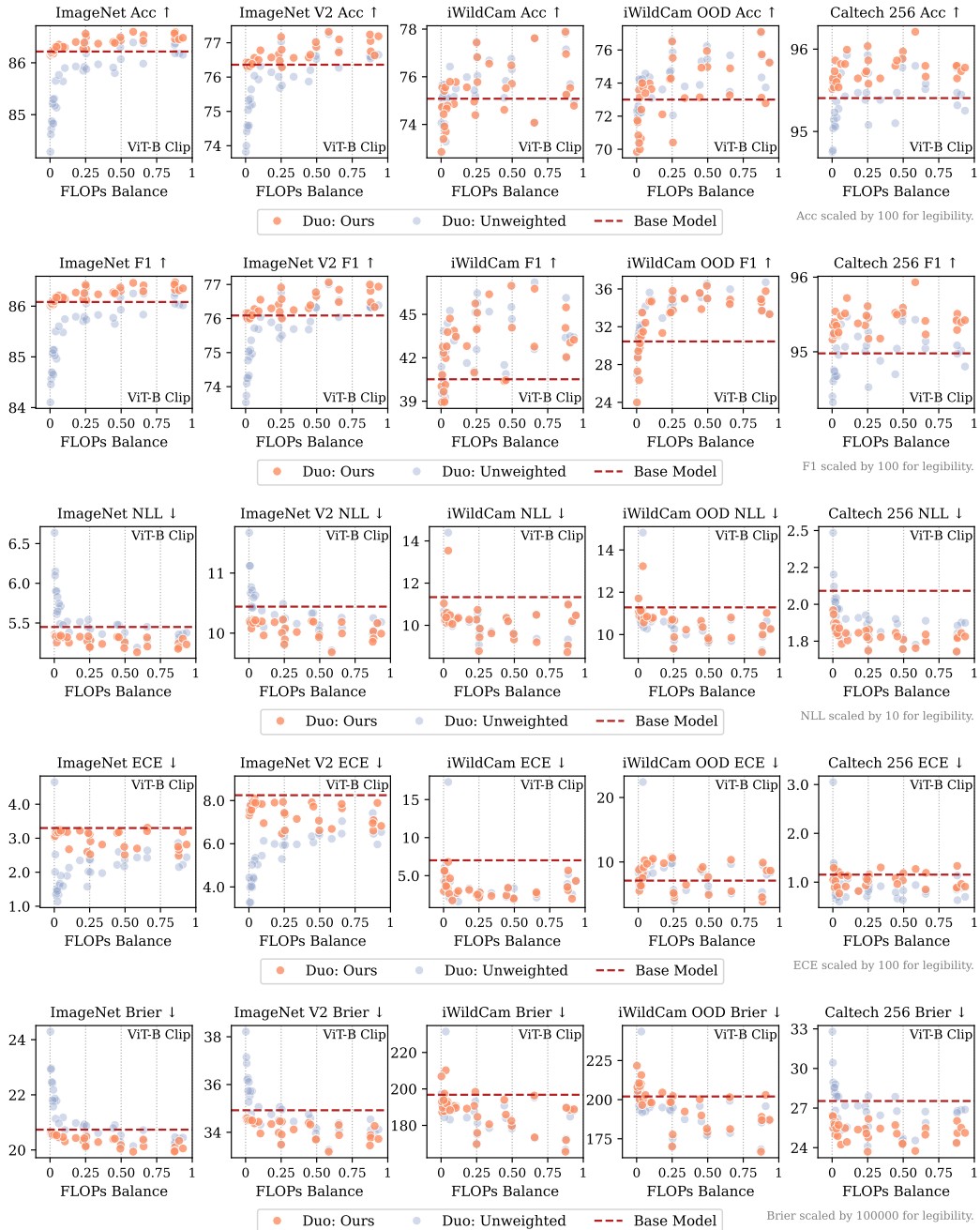

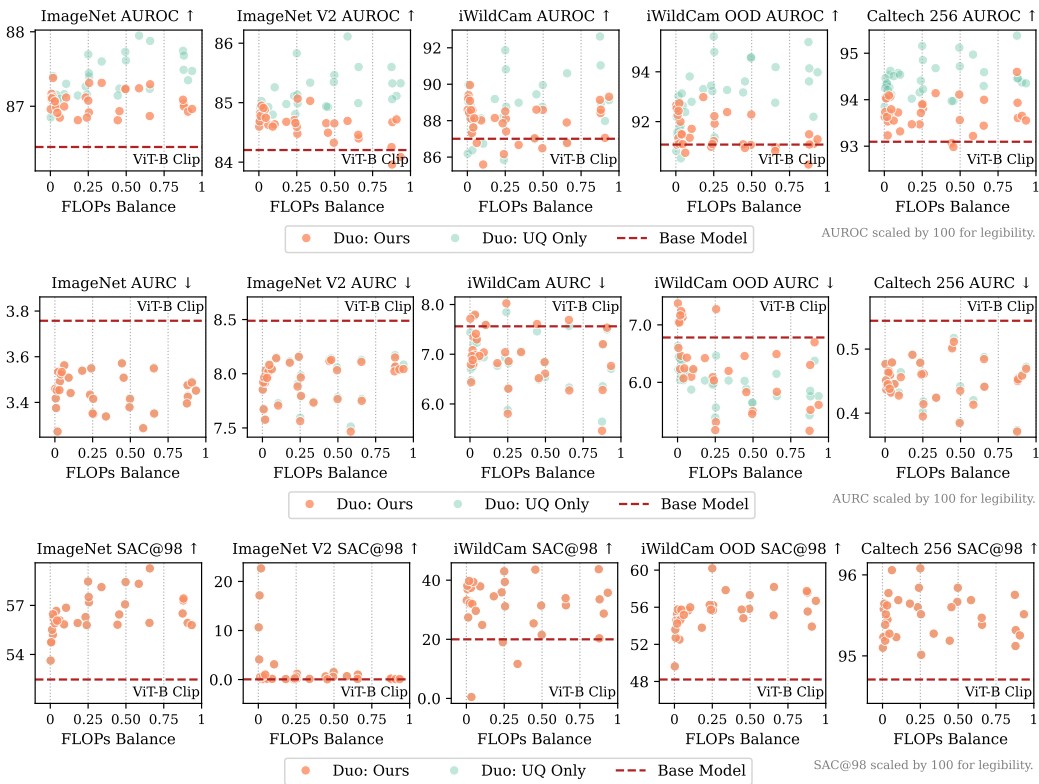

Figure 11: Accuracy, F1, NLL, ECE, Brier, AUROC, AURC, SAC@98 for all $f_{\mathrm{large}}$ = ViT-B Duos.

**Swin-V2-B Pretrained on ImageNet1K** is benchmarked as $f_{\text{large}}$ on all five datasets.

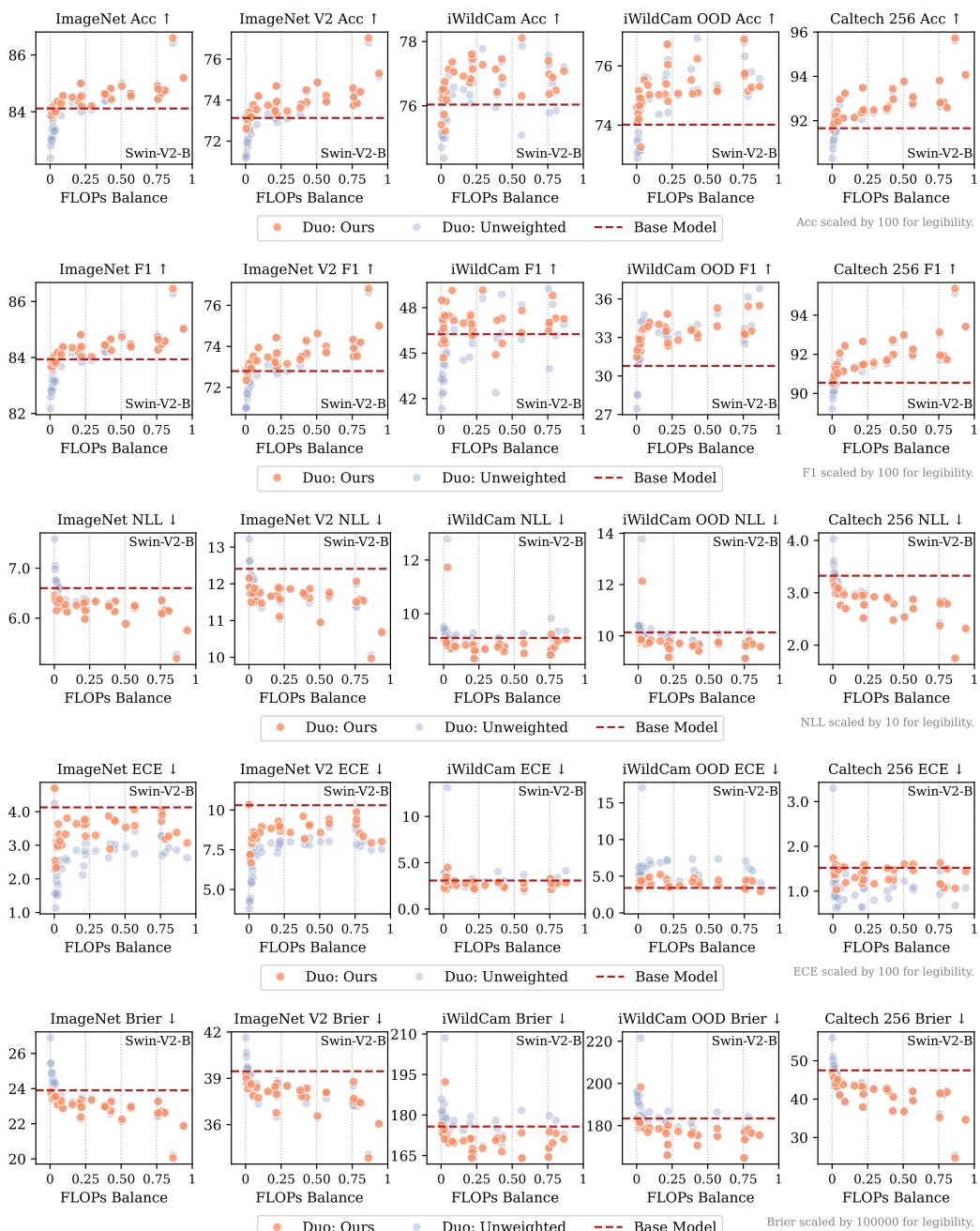

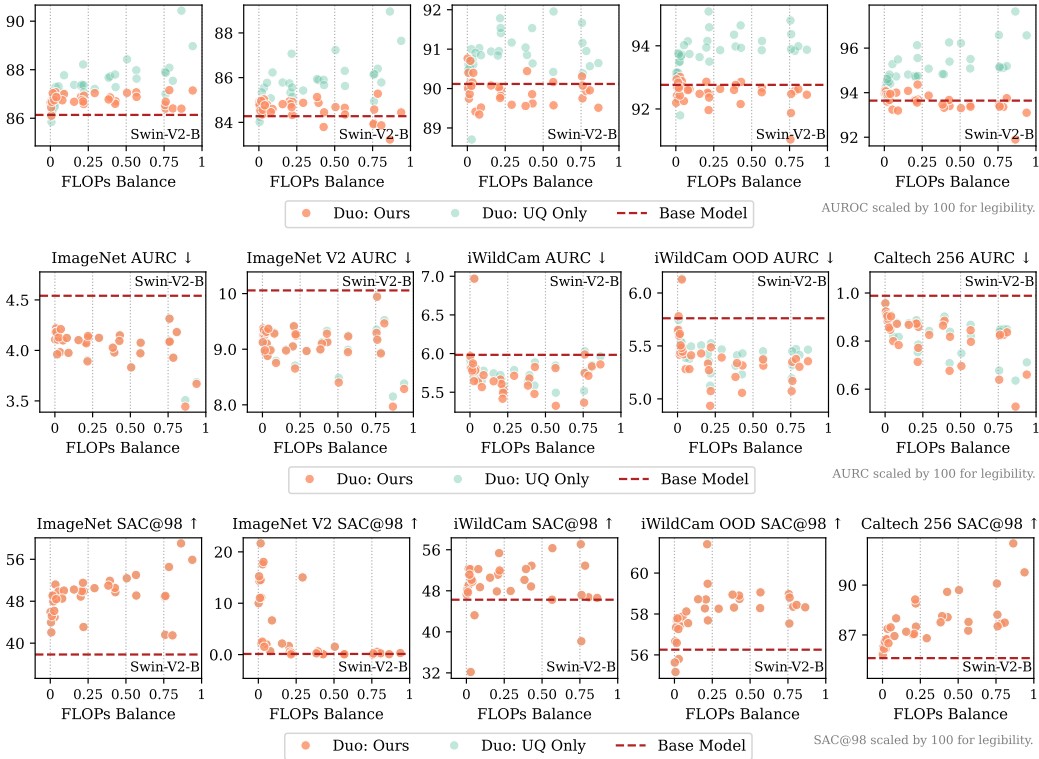

Figure 12: Accuracy, F1, NLL, ECE, Brier, AUROC, AURC, SAC@98 for all $f_{\text{large}}$ = Swin-V2-B Duos.

**EfficientNet-V2-L Pretrained on LAION-2B** is benchmarked as $f_{\text{large}}$ on all five datasets.

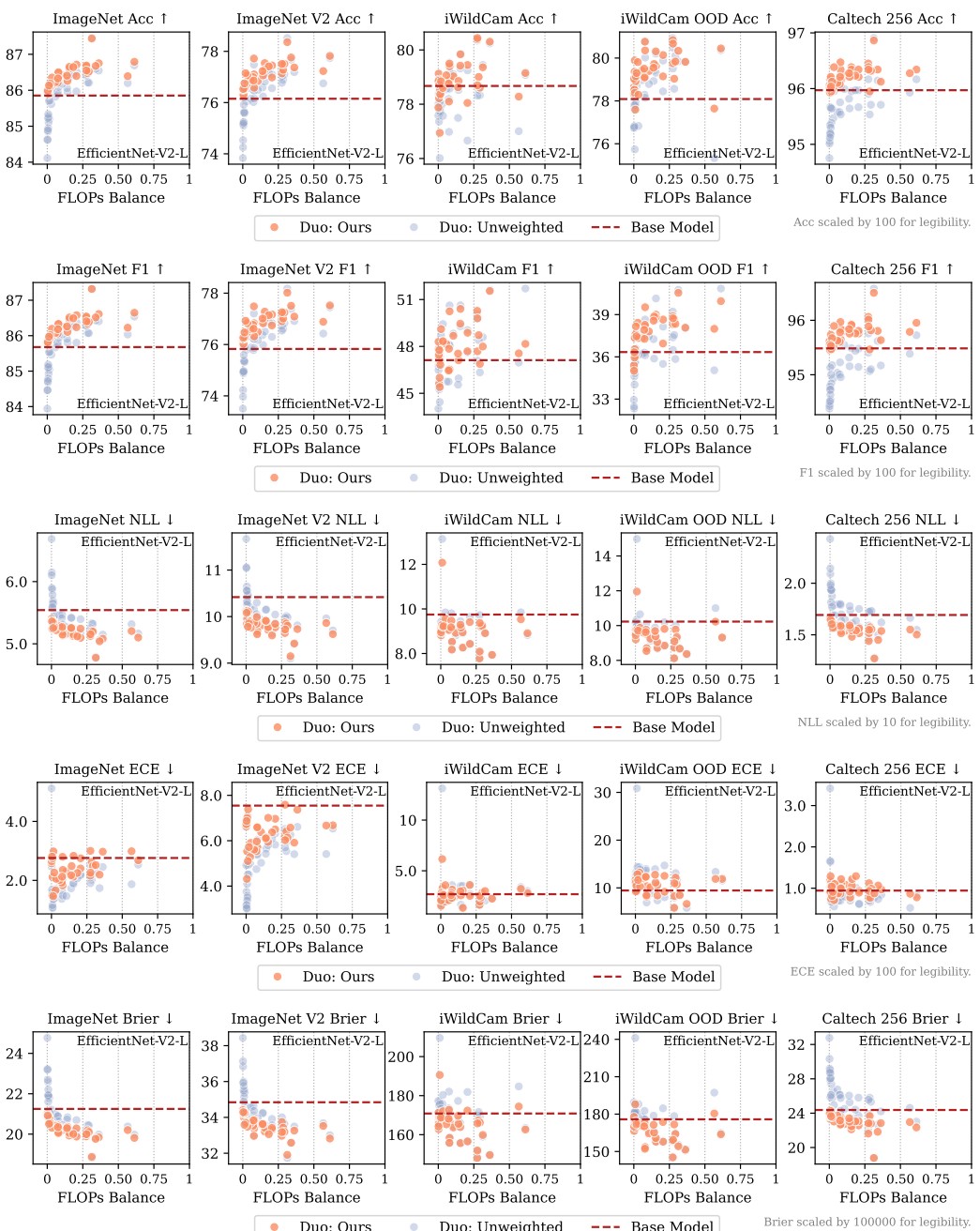

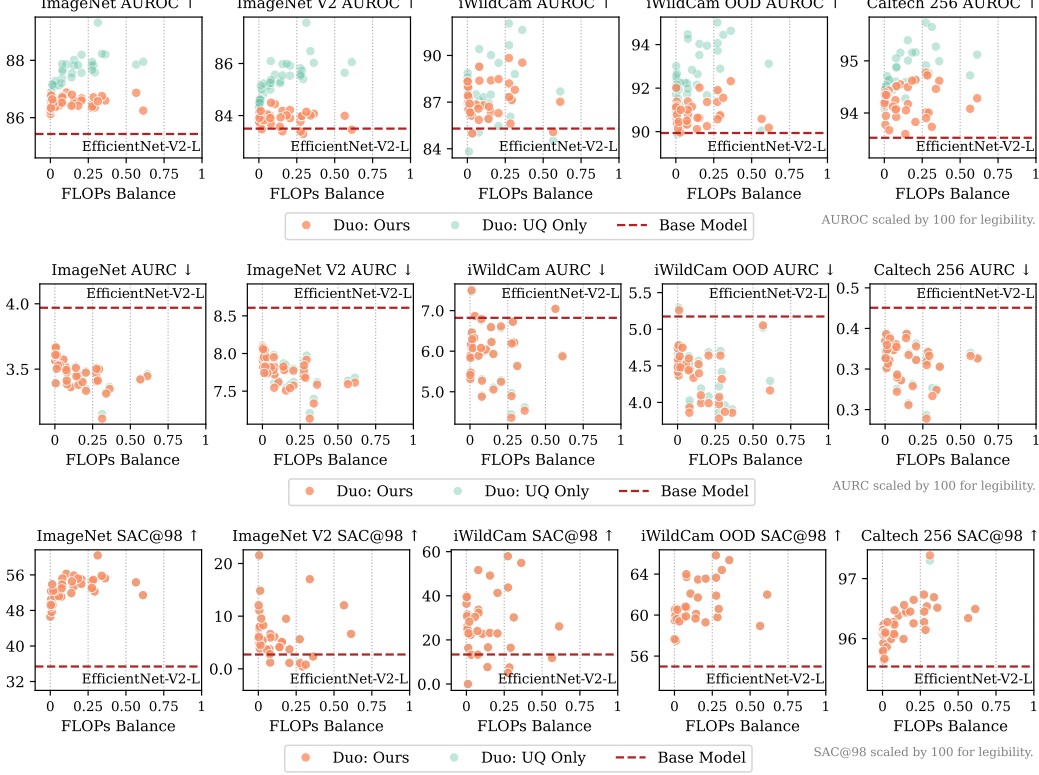

Figure 13: Accuracy, F1, NLL, ECE, Brier, AUROC, AURC, SAC@98 for all $f_{\text{large}}$ = EfficientNet-V2-L Duos.

**ConvNeXt-Base Pretrained on ImageNet-1K** is benchmarked as $f_{\text{large}}$ on all five datasets.

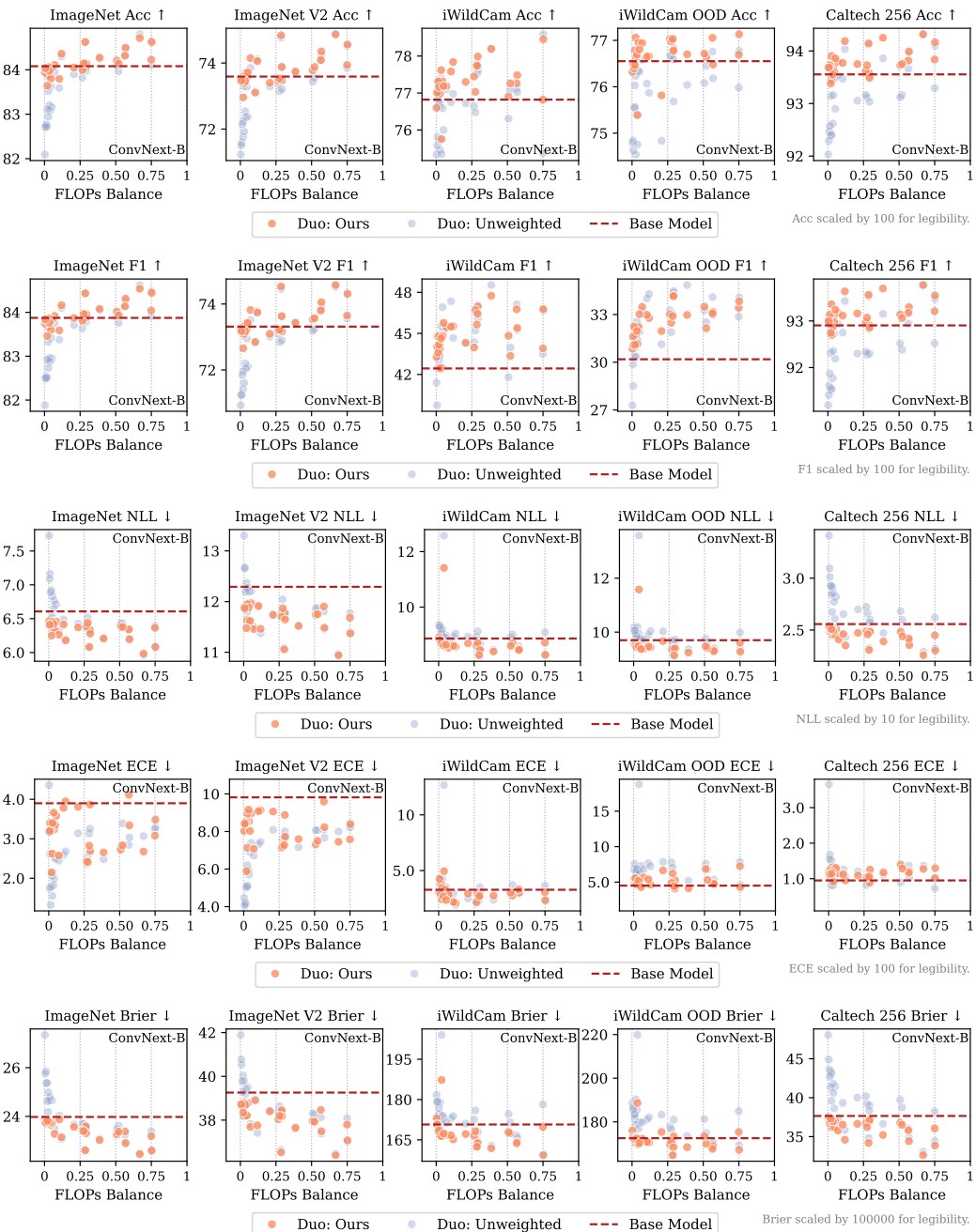

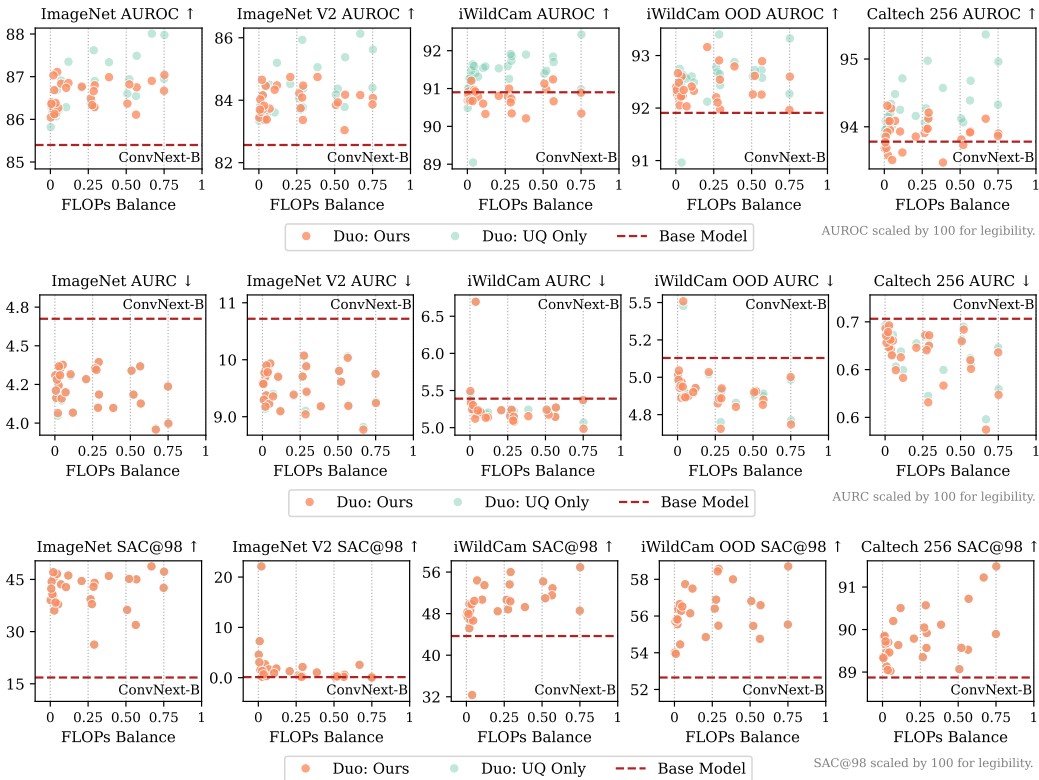

Figure 14: Accuracy, F1, NLL, ECE, Brier, AUROC, AURC, SAC@98 for all $f_{\text{large}} =$ ConvNeXt-Base Duos.

