# OpenReview forum: "Asymmetric Duos: Sidekicks Improve Uncertainty"
_NeurIPS.cc/2025/Conference — NeurIPS 2025 spotlight_

### Official Review · Reviewer_VDfk · 2025-06-12

**Clarity:** 4
**Significance:** 2
**Originality:** 2
**Rating:** 5
**Confidence:** 4

**Summary:**

The paper proposes to build an ensemble of n=2 models, especially when the second model is smaller than the first. The models are combined on the logit level by learning mixture weights (= temperatures) based on validation log likelihood. The authors find that these “Duos” give increased performance, both in terms of accuracy and uncertainty quantification, and that the second model does not have to be big for this to be true. Indeed, if the second model has about 25% of the FLOPs of the first model, the performance already saturates. This indicates that model ensembles can be more efficient than the popular default n=10 deep ensembles.

**Questions:**

See the request for other ensemble baselines.

Since you mix the two models on the logit level (=unnormalized), it could be that one model generally has logit vectors with bigger norms than another, such that there is a confounding factor besides the mixture temperatures. I do not expect this to happen systematically, but could you check whether this is an issue, just in case?

**Ethical Concerns:**

["NO or VERY MINOR ethics concerns only"]

**Final Justification:**

The rebuttal has addressed the main point of critique (of me and other reviewers) by adding the deep ensemble results. As auch, I believe the paper is now well-rounded off and technically solid. Although not award-winning, I think the paper should be accepted, and because I believe rebuttal periods should be used to get out of borderline scores, I update to a score of 5.

**Limitations:**

The authors describe limitations and related works that are close to their results.

**Paper Formatting Concerns:**

None.

**Quality:**

4

**Strengths And Weaknesses:**

## Strengths

* The paper is well written and section connect to each other in a logical manner
* The paper pays attention to detail, such as reporting number of parameters + FLOPS + throughput to give a wholistic picture, adding ablations on metrics and models in the appendices, and making sure comparisons are fair (for example using the UQ-only Duo when comparing AURC)
* The analysis is very deep in terms of metrics (accuracy, logl, Brier score, ECE, AUROC, AURC, SAC). All of these metrics are grouped nicely and put into perspective correctly, I am not missing anything here.
* The paper is reproducible. The appendix contains an in-depth description of all hyperparameters and code that reproduces Figure 1-14 is provided.

## Weaknesses

### Major weaknesses (that influence my score)

* The paper only compares duos to single models as a baseline. It would be interesting to compare to deep ensembles of different sizes, if not further modern ensembling approaches. Currently, the paper only supports the hypothesis “Yes, adding a second model improves performance and UQ of the duo, even if the second model is small.”, but comparing to ensembles could help judge if the improvement is already saturating to the performance of the ensemble (showing that deep ensembles are unnecessarily big), or how big the gap is. I would be happy to see experiments on this in the rebuttal. Regardless of whether the outcome is “positive” or not, I think it would strengthen the paper.
* The novelty of the paper is incremental. In essence, the main message is “Adding a second model helps improve performance slightly, and this second model can be smaller (although bigger = better)”. Since ensembling is already well-known, and also n=2 ensembles were explored in literature, the methodological and insight contribution is limited, and it is also limited in scope to image classification, which is a topic that ages currently.

### Other notes that did not influence my score and need no rebuttal but could help improve the camera-ready:

* Caption of Figure 2, I would replace “littler” with “smaller”
* “Randomly assigned uncertainty measure would yield an AUROC score of 0.5“ → might want to rewrite to either a measure / estimator that randomly assigns, or randomly assigned estimates. A randomly assigned measure is somewhat unorthodox
* “We note that the AUROC metric couples accuracy and uncertainty quantification“ I do not believe this is true. The AUROC (Section 5.2) is independent of the marginal of the binary variable, which in your case is the accuracy. (the AURC in section 5.3 does conflate them)

## Justification for score

The paper is exceptionally well executed both in experiments and in writing. However, its novelty is somewhat incremental. Since I generally support well-executed papers, even if not exactly novel, I recommend to accept the paper. However, I am looking forward to experiments with deep-ensemble and other ensembling baselines, which I believe are somewhat missing with the papers scope of showing that ensembles don’t need to be big. I am willing to increase my score if these experiments can be provided, whether the outcome is “negative” or “positive” for the papers message.

---

> ### Author Rebuttal · Authors · 2025-07-30
>
> # Author Response to Reviewer VDfk
>
> We thank the reviewer for the thoughtful review and the constructive feedback.
>
> ---
>
> **[W1] Direct Comparison between Asymmetric Duos (AD) and Deep Ensembles (DE).**
> We thank the reviewer for highlighting this important comparison. Here we directly compare ADs and ResNet-50 DEs on ImageNet while controlling for total FLOPS. In short, while Deep Ensembles show diminishing returns as the number of members ($m$) increases, Duos continue to improve in accuracy and uncertainty quantification with higher FLOPS. This makes Duos more effective under high compute budgets.
>
> We provide the results and experimental details below:
>
> | Method      | FLOPS(G)     | Acc↑       | Brier↓    | NLL↓       | ECE↓       | AUROC↑    | AURC↓    |
> |-------------|---------------|------------|-----------|------------|------------|-----------|-----------|
> | Base Model  | 4.1           | 79.8±0.5   | 3.3±0.5   | 102.6±14.1 | 16.8±9.6   | 85.0±1.9  | 6.4±0.8   |
> | AD[1,2)     | [4.1,8.2)    | 81.2±0.8   | 2.7±0.1   | 74.1±4.0   | 2.5±0.8    | 86.9±0.7  | 5.1±0.3   |
> | DE(m=2)     | 8.2           | 81.2±0.3   | 2.9±0.2   | 88.8±9.7   | 12.4±7.5   | 86.4±1.0  | 5.3±0.5   |
> | AD[2,3)     | [8.2,12.3)   | 82.3±0.6   | 2.5±0.1   | 70.2±3.1   | 3.0±0.8    | 87.1±0.7  | 4.7±0.3   |
> | DE(m=3)     | 12.3          | 81.7±0.2   | 2.8±0.2   | 83.5±9.4   | 9.4±7.9    | 86.8±0.8  | 4.9±0.4   |
> | ...         |               |            |           |            |            |           |           |
> | AD[4,5)     | [16.4,20.5)  | 84.8±1.2   | 2.2±0.1   | 59.4±5.4   | 2.9±0.7    | 86.9±0.7  | 3.9±0.4   |
> | DE(m=5)     | 20.5          | 82.0±0.2   | 2.7±0.2   | 79.0±7.7   | 6.9±6.5    | 87.1±0.6  | 4.6±0.3   |
> | ...         |               |            |           |            |            |           |           |
> | AD[7,8)     | [28.7,32.8)  | 86.3±0.2   | 2.0±0.0   | 53.0±0.9   | 3.0±0.4    | 87.3±0.3  | 3.4±0.1   |
> | DE(m=8)     | 32.8          | 82.3±0.1   | 2.6±0.1   | 76.8±3.8   | 5.2±3.7    | 87.3±0.2  | 4.5±0.1   |
>
>
> This comparison highlights the superior scalability of Asymmetric Duos under FLOPS budgets.
> AD$[m,m+1)$ denotes Asymmetric Duos with total FLOPS within $m$ to $m+1$ times of that of a Base ResNet50 Model. Note that all metrics values were scaled by $10^{2}$ for better presentation, with the exception of Brier score which was scaled by $10^{4}$. DEs of sizes $m=2$ to $8$ are formed by 32 random subsets of 10 ResNet-50 checkpoints from timm; ADs compose a ResNet50/ResNet101/ResNet152/ConvNeXt-Base/ViT-B base model with a sidekick model with FLOPS Balances $\geq0.1$.
>
> We also repeated this analysis controlling for parameter count instead of FLOPS, and observed similar trends. These results and their visualizations will be included in the Appendix in our revision. These results and their visualizations will be included in the Appendix in our revision. We hope our findings on Asymmetric Duos would spark broader interest in model combination methods, including homogeneous DEs, and this direct comparison would contribute to a better understanding of when and how to apply each approach effectively.
>
> ---
>
> **[W2] Incremental Novelty.**
> We appreciate the reviewer's perspective, but note the empirical novelty of our heterogenous Duos, that can effectively combine two fine-tuned networks of vastly different sizes without joint fine-tuning. Asymmetric Duos are uniquely poised to be highly compatible with today's large pretrained models and fine-tuning workflow, and are able to "interpolate" the accuracy-FLOPs trade off by adding fractional additional computation w.r.t. to the base model, introducing finer control over the scaling laws at minimal computational cost.
>
> ---
>
> **[Q1] Scales of Un-normalized Logit Vectors.**
> This point on a potential scale mismatch of raw logits is easy to overlook, we thank the reviewer for the careful observation.
>
> You're right that naively averaging logits from heterogeneous models can introduce bias due to norm mismatches; an issue we also observed in the Unweighted (ablation) Duos. Our AD strategy specifically addresses this through temperature-based logit scaling. Since scaling logits by a temperature also scales their outputs' norm linearly, the learned temperatures implicitly adjust the scale between model outputs. For instance, on the ImageNet validation set, EfficientNet-V2-L (145M parameters) and EfficientNet-B0 (5M parameters) have average logit vector l2 norms of 24.50 and 33.07, respectively. Our learned temperatures (1.64 for V2-L and 0.86 for B0) adjust these to 40.21 and 28.31, effectively up-weighting the base model’s contribution.
> We observe similar results for all Duos on all datasets, and will include this discussion in the revision to alleviate concerns around this point.
>
> ---
>
> We thank the reviewer again for the thoughtful feedback. We hope our responses adequately address the concerns raised and would be happy to discuss further during the reviewer–author discussion period.

---

> ### Author Response · Authors · 2025-08-04
> **Direct Comparison of Asymmetric Duos and Deep Ensembles with Fixed Base Model**
>
> We thank the reviewer for their quick and thoughtful response. We're encouraged that our rebuttal helped reinforce the reviewer's positive assessment of our work.
>
> We also appreciate the clarification regarding the Deep Ensemble comparison. We agree that this base-model-controlled setup is a meaningful and practical complement to our FLOPs-controlled comparison. Both offer useful perspectives, and we now include results under the reviewer’s requested setting using ResNet50 (RN50) for all Ensembles members and Duos $f_{\text{large}}$.
>
> | Method     | Acc↑      | Brier↓   | NLL↓        | ECE↓      | AUROC↑    | AURC↓    |
> |:-----------|:-----------|:----------|:-------------|:-----------|:-----------|:----------|
> | RN50 (Temperature Scaled) | 79.82±0.51 | 2.90±0.05 | 84.06±3.38  | 3.39±1.39  | 86.62±0.49 | 5.72±0.38 |
> | RN50+MN.75 AD (FB=0.05) | 79.88±0.42 | 2.87±0.05 | 80.45±2.17 | 3.25±0.61 | 87.11±0.29 | 5.42±0.19 |
> | RN50+SN2.0x AD (FB=0.14) | 80.32±0.32 | 2.81±0.04 | 78.93±1.78 | 2.62±0.76 | 87.00±0.27 | 5.34±0.17 |
> | RN50+ENB2 AD (FB=0.27) | 81.87±0.19 | 2.60±0.03 | 71.47±0.84 | 2.80±0.48 | 87.15±0.26 | 4.80±0.17 |
> | RN50 DEs (m=2)   | 81.12±0.29 | 2.72±0.04 | 77.12±1.90 | 3.54±1.33 | 87.19±0.30 | 4.99±0.24 |
> |...| | | | | | |
> | RN50 DEs (m=5)   | 81.97±0.13 | 2.61±0.01 | 73.16±0.90 | 3.79±0.76 | 87.62±0.11 | 4.48±0.09 |
>
> Here, Deep Ensembles follow the “pool-then-calibrate” strategy from [Rahaman2021]. Since our temperature-based Duo aggregation inherently performs calibration, this enables a fairer comparison on NLL, Brier, and ECE. All ensembles are drawn from random subsets of 10 independently trained ResNet50 (RN50) checkpoints (from both timm and torchvision). As requested, Duos in this setting are $f_{\text{large}}$-controlled: each Duo shares a fixed RN50 as $f_{\text{large}}$, and varies its sidekick among MNASNet0.75 (MN.75), ShuffleNet V2 2.0× (SN2.0x), and EfficientNet-B2 (ENB2), with FLOPs Balances (FB) of 0.05, 0.14, and 0.27, respectively.
>
> Despite their lower compute, these asymmetric RN50 Duos quickly approach performance of larger, homogeneous Deep Ensembles (e.g., $m=5$ RN50s) across most metrics. We attribute this to architectural diversity from heterogeneity [Gontijo-Lopes2022] and Duos' simple but effective temperature-based aggregation. Together, these allow RN50-based Duos to realize a large portion of the performance gains of full $m=5$ Deep Ensembles at a fraction of the cost.
>
> We hope this base-model-controlled comparison directly addresses the reviewer’s concern. We have updated the manuscript to include both the FLOPs-controlled and $f_{\text{large}}$-controlled Duo comparisons in a new Appendix section titled “Evaluating Asymmetric Duos against Deep Ensembles”. There, we note that ensembling remains a strong approach in uncertainty-critical scenarios, and Duos offer an alternative that retains much of the accuracy and UQ benefits. Relevant pointers have been added to both the main Experiments and Discussion sections.
>
> Note that we are including the same response for Reviewer ZFrX, who raised a similar concern. We appreciate this alignment, as it clearly signals the value of including this comparison in our paper.We believe this addition makes our analysis more thorough and helps better position Duos among other methods for practitioners, as the reviewer suggested. We thank the reviewer for this thoughtful suggestion on direct comparisons between DEs and $f_{\text{large}}$-controlled ADs.
>
>
> ---
>
> **References**
>
> [Rahaman2021] Rahul Rahaman and Alexandre H Thiery. *Uncertainty quantification and deep ensembles.* Proceedings of the 35th International Conference on Neural Information Processing Systems,
> 2021.
>
> [Gontijo-Lopes2022] Raphael Gontijo-Lopes, Yann Dauphin, and Ekin Dogus Cubuk. *No one representation to rule them all: Overlapping features of training methods.* International Conference on Learning Representations, 2022.

---

> ### Comment · Reviewer_VDfk · 2025-08-05
>
> I would like to thank the authors for the results, which I believe make the experimental evaluation more complete now. The findings look solid and allow to place the proposed assymetric duos nicely in the landscape of other models. I appreciate that you were able to deliver these results within the discussion period, good work!
>
> My questions and concerns are addressed, I will continue to monitor the other reviewer discussions to make sure I did not miss anything, but will then update my final score.

---

### Official Review · Reviewer_ZFrX · 2025-06-14

**Clarity:** 3
**Significance:** 2
**Originality:** 3
**Rating:** 5
**Confidence:** 4

**Summary:**

The paper introduces Asymmetric Duos, a practical and cost-effective method for uncertainty quantification (UQ) that leverages an ensemble of two models: a large model and a considerably smaller "sidekick." The final prediction is obtained via a simple learned temperature-weighted aggregation of both outputs in logit space. The approach is evaluated in the image classification setting and is shown to improve predictive accuracy and UQ with negligible overhead, making it particularly attractive in large-scale or fine-tuning regimes.

**Questions:**

- **[Q1]** How do the learned temperatures behave in practice? Do they consistently emphasize a single model or does the dominance of one model balance out with FLOPS? Are there "mode-collapses"? Could the authors quantify how often the sidekick model meaningfully changes the prediction, e.g., via majority class swaps?


- **[Q2]** How does the performance (accuracy/UQ) of Asymmetric Duos compare against a standard Deep Ensemble with 5–10 models on smaller datasets (and other scalable UQ methods)? See [W1].


- **[Q3]**:  Do the authors see potential connections between the proposed Duos approach and Bayesian Deep Learning—particularly sampling-based inference—which also constructs ensembles of model instances to approximate the Bayesian model average (BMA)? Would you then expect a BMA composed of posterior samples from asymmetric architectures to perform well?


$\rightarrow$ If the authors can satisfactorily address the raised weaknesses and questions, in particular  W1/Q2, I would be inclined to raise my score to an accept.

**Ethical Concerns:**

["NO or VERY MINOR ethics concerns only"]

**Final Justification:**

I recommend acceptance.

My initial score was borderline, but my concerns have been, for the most part, addressed through a productive discussion and new experiments provided by the authors.

The rebuttal successfully resolved the main weaknesses regarding the experimental evaluation [W1, W2] by providing new, crucial comparisons against both standard Deep Ensembles and other scalable UQ methods. This new evidence significantly strengthens the paper's claims and contextualizes its contribution.

My final recommendation is contingent on the authors integrating all new results and the related discussions addressing the other raised weaknesses/questions into the final manuscript.

**Limitations:**

yes

**Paper Formatting Concerns:**

None.

**Quality:**

3

**Strengths And Weaknesses:**

## Strengths
- **[S1]** The approach is simple and practical, offering improvements in predictive accuracy and UQ with very low computational overhead.
- **[S2]** Applicable in standard fine-tuning scenarios and scales well to large architectures.
- **[S3]** The empirical evaluation is thorough within the considered setup, and the paper is clearly written and easy to follow.

## Weaknesses

**Major:**

**[W1]** The study focuses exclusively on very large architectures. However, it is known from theory and practice that deep ensembles of moderate size (e.g., 5–10 models) can offer significant improvements and can themselves even be improved significantly (cf. [1,2,3]). It would be insightful to compare Asymmetric Duos against larger deep ensemble sizes - potentially even other UQ methods see W2 - on smaller-scale problems where these are computationally feasible. This would allow a rough assessment of how large the *cost-quality trade-off* is between DEs with usual ensemble sizes and Asymmetric Duos (and potentially other competing UQ methods).

**Minor:**
- **[W2]** No comparison is made with scalable approximate Bayesian methods like IVON [4] or Laplace Redux [5]. While not critical, such a comparison could better contextualize the proposed method in the UQ landscape.
- **[W3]** The aggregation scheme is simplistic—fixed temperature scaling. The paper lacks theoretical development or empirical exploration of even simple data-dependent or more complex adaptive combination strategies. As the simple one seems to work already well this point is minor. A closer look at potentially persistent performance gaps with larger ensemble sizes (see W1) could help in assessing how much room for improvement from more evolved aggregations can be anticipated.
- **[W4]** The discussion of individual results and metrics is occasionally redundant and could be streamlined for conciseness.
## References

- [1] Wang, Y., & Wang, X. (2025). Agree to Disagree: Demystifying Homogeneous Deep Ensembles through Distributional Equivalence. In *The Thirteenth International Conference on Learning Representations*.
- [2] Wild, V. D., Ghalebikesabi, S., Sejdinovic, D., & Knoblauch, J. (2024). A Rigorous Link between Deep Ensembles and (Variational) Bayesian Methods. *Advances in Neural Information Processing Systems*, 36.
- [3] Sommer, E., Robnik, J., Nozadze, G., Seljak, U., & Rügamer, D. (2025). Microcanonical Langevin Ensembles: Advancing the Sampling of Bayesian Neural Networks. In *The Thirteenth International Conference on Learning Representations*.
- [4] Shen, Y., Daheim, N., Cong, B., Nickl, P., Marconi, G. M., Raoul, B. C. E. M., Yokota, R., Gurevych, I., Cremers, D., Khan, M. E., & Möllenhoff, T. (2024). Variational Learning is Effective for Large Deep Networks. In *Proceedings of the 41st International Conference on Machine Learning*, *Proceedings of Machine Learning Research*, 235:44665–44686. Available from: https://proceedings.mlr.press/v235/shen24b.html.
- [5] Daxberger, E., Kristiadi, A., Immer, A., Eschenhagen, R., Bauer, M., & Hennig, P. (2021). Laplace Redux – Effortless Bayesian Deep Learning. In *35th Conference on Neural Information Processing Systems (NeurIPS 2021)*.

---

> ### Author Rebuttal · Authors · 2025-07-30
>
> # Author Response to Reviewer ZFrX
>
> ---
>
> We thank the reviewer for the thoughtful review and the insightful suggestions.
>
> **[W1&Q2] Direct Comparison between Asymmetric Duos (AD) and Deep Ensembles (DE).**
> We thank the reviewer for highlighting this important comparison. Here we directly compare ADs and ResNet-50 DEs on ImageNet while controlling for total FLOPS. In short, while Deep Ensembles show diminishing returns as the number of members ($m$) increases, Duos continue to improve in accuracy and uncertainty quantification with higher FLOPS. This makes Duos more effective under high compute budgets. We provide the results and experimental details below:
>
> | Method      | FLOPS(G)     | Acc↑       | Brier↓    | NLL↓       | ECE↓       | AUROC↑    | AURC↓    |
> |-------------|---------------|------------|-----------|------------|------------|-----------|-----------|
> | Base Model  | 4.1           | 79.8±0.5   | 3.3±0.5   | 102.6±14.1 | 16.8±9.6   | 85.0±1.9  | 6.4±0.8   |
> | AD[1,2)     | [4.1,8.2)    | 81.2±0.8   | 2.7±0.1   | 74.1±4.0   | 2.5±0.8    | 86.9±0.7  | 5.1±0.3   |
> | DE(m=2)     | 8.2           | 81.2±0.3   | 2.9±0.2   | 88.8±9.7   | 12.4±7.5   | 86.4±1.0  | 5.3±0.5   |
> | AD[2,3)     | [8.2,12.3)   | 82.3±0.6   | 2.5±0.1   | 70.2±3.1   | 3.0±0.8    | 87.1±0.7  | 4.7±0.3   |
> | DE(m=3)     | 12.3          | 81.7±0.2   | 2.8±0.2   | 83.5±9.4   | 9.4±7.9    | 86.8±0.8  | 4.9±0.4   |
> | ...         |               |            |           |            |            |           |           |
> | AD[4,5)     | [16.4,20.5)  | 84.8±1.2   | 2.2±0.1   | 59.4±5.4   | 2.9±0.7    | 86.9±0.7  | 3.9±0.4   |
> | DE(m=5)     | 20.5          | 82.0±0.2   | 2.7±0.2   | 79.0±7.7   | 6.9±6.5    | 87.1±0.6  | 4.6±0.3   |
> | ...         |               |            |           |            |            |           |           |
> | AD[7,8)     | [28.7,32.8)  | 86.3±0.2   | 2.0±0.0   | 53.0±0.9   | 3.0±0.4    | 87.3±0.3  | 3.4±0.1   |
> | DE(m=8)     | 32.8          | 82.3±0.1   | 2.6±0.1   | 76.8±3.8   | 5.2±3.7    | 87.3±0.2  | 4.5±0.1   |
>
>
> This comparison highlights the superior scalability of Asymmetric Duos under FLOPS budgets.
> AD$[m,m+1)$ denotes Asymmetric Duos with total FLOPS within $m$ to $m+1$ times of that of a Base ResNet50 Model. Note that all metrics values were scaled by $10^{2}$ for better presentation, with the exception of Brier score which was scaled by $10^{4}$. DEs of sizes $m=2$ to $8$ are formed by 32 random subsets of 10 ResNet-50 checkpoints from timm; ADs compose a ResNet50/ResNet101/ResNet152/ConvNeXt-Base/ViT-B base model with a sidekick model with FLOPS Balances $\geq0.1$.
>
> We also repeated this analysis controlling for parameter count instead of FLOPS and observed similar trends. These results and their visualizations will be included in the Appendix in our revision. We hope these findings will spark broader interest in methods for both homogenous combination (ensembles) and heterogeneous combination (asymmetric Duos) and contribute to a better understanding of when and how to apply each approach effectively.
>
> ---
>
> **[W2] Scalable Bayesian Methods.**
> We thank the reviewer for making thoughtful and specific suggestions on scalable bayesian methods.
>
> **Improved Variational Online Newton (IVON)**
>
> We benchmark Asymmetric Duos against a ResNet50 fine-tuned with IVON optimizer on Caltech256.
>
> | Method                    | Acc↑       | Brier↓     | NLL↓       | ECE↓       | AUROC↑     | AURC↓     |
> |---------------------------|------------|------------|------------|------------|------------|-----------|
> | Baseline ResNet50         | 87.72      | 7.60       | 67.90      | 7.56       | 91.81      | 1.92      |
> | Temp-Scaled ResNet50      | 87.72      | 6.88       | 49.54      | 1.22       | 91.44      | 2.01      |
> | IVON (deterministic)      | 87.98      | 6.63       | 45.33      | 1.89       | 91.92      | 1.83      |
> | Asymmetric Duos [1,2)     | 89.75±1.17 | 5.80±0.60  | 40.73±4.56 | 0.97±0.28  | 91.58±0.53 | 1.52±0.18 |
> | IVON (2 samples)          | 87.23±0.33 | 7.21±0.09  | 49.44±0.58 | 1.59±0.26  | 91.33±0.27 | 2.09±0.06 |
> | IVON (4 samples)          | 87.98±0.17 | 6.91±0.05  | 47.08±0.37 | 2.60±0.22  | 91.40±0.27 | 1.89±0.05 |
> | IVON (8 samples)          | 88.19±0.16 | 6.77±0.02  | 45.88±0.22 | 3.10±0.20  | 91.65±0.28 | 1.81±0.02 |
> | IVON (16 samples)         | 88.41±0.17 | 6.72±0.01  | 45.52±0.14 | 3.36±0.19  | 91.47±0.29 | 1.78±0.01 |
> | IVON (32 samples)         | 88.43±0.13 | 6.68±0.02  | 45.17±0.15 | 3.46±0.15  | 91.58±0.10 | 1.77±0.02 |
>
>
> IVON works well out of the box, outperforming standard AdamW with minimal training overhead.
> Its sampling-based inference further improved over the deterministic version, though like Deep Ensembles, it introduces an $m\times$ test-time cost with diminishing returns.
> In contrast, Asymmetric Duos built on ResNet-50 achieve better performance across all metrics with only a fractional increase in compute.
> Furthermore, similar to the compatibility shown between soup and Asymmetric Duos in Section 5.5, IVON, as a great plug-and-use uncertainty-aware optimizer, would fit nicely with our Asymmetric Duos framework.
>
>
> **Laplace Approximation**
>
> We also compare Laplace Approximation (LA) with Asymmetric Duos. Due to LA's higher computational and memory usage when applied to architectures with high last-layer dimension, we perform this analysis on Caltech256 using a Swin-T model.
>
> | Method                       | Acc↑      | F1↑       | Brier↓   | NLL↓      | ECE↓     | AUROC↑   | AURC↓    | SAC@98↑   |
> |:-----------------------------|:---------|:---------|:--------|:---------|:--------|:-----------|:--------|:---------|
> | Swin-T baseline              | 90.2     | 89.2     | 5.7     | 41.3     | 4.0     | 92.8       | 1.3     | 80.3     |
> | Swin-T temp-scaled           | 90.2     | 89.2     | 5.5     | 37.5     | 1.7     | 92.9       | 1.3     | 79.8     |
> | Swin-T Laplace Approximation | 90.7     | 89.6     | 5.7     | 47.4     | 5.4     | 93.3       | 1.2     | 81.3     |
> | Swin-T Asymmetric Duos (<0.3 FLOPS Balance)              | 91.5±0.3 | 90.6±0.3 | 4.9±0.2 | 33.2±1.4 | 1.4±0.2 | 93.0±0.3   | 1.0±0.1 | 83.4±1.1 |
>
> We observe that, similar to IVON, Laplace Approximation (last-layer-only & kron & probit as recommended) reliably improves over the base model. However, it falls short in absolute performance gains compared to Asymmetric Duos across most metrics. Notably, LA can also be incorporated into any Duo, further demonstrating the generality and practicality of the Asymmetric Duos framework.
>
> ---
>
> **[W3] More Asymmetric Duos Aggregation.**
> We thank the reviewer for recognizing that the simple fixed temperature technique is effective. We argue this simplicity is a strength: if we select temperature weights to maximize validation accuracy, Duos necessarily outperform the large model on validation data (the large model is simply a Duo with weights [1,0]). Given the low VC dimension of the space of possible temperature weightings (≤3), standard learning theory provides O($\sqrt{1/|\mathrm{val_set}|}$) generalization bounds on accuracy improvement with small constants. We will include a derivation of this bound in revision (and we can provide details to the reviewers if needed, though it follows from standard learning theory). Second, the efficacy of such a simple approach is itself a notable empirical contribution. While established work suggests that ensembling models of different performance levels typically yields degradations [Opitz1999; Wood2023], our finding that asymmetric models can be effectively combined with simple temperature weighting should surprise the community. Again, we will include this discussion in the revision.
> While data-dependent mechanisms could yield further gains, they would sacrifice these theoretical guarantees and require additional training overhead. Nevertheless, we believe pursuing such "Dynamic Duos" is a fruitful direction for future work.
>
> ---
>
> **[Q1] Temperature behaviors in practice.**
> Learned temperatures tend to emphasize the base model in low-FLOPS-balance regimes and gradually increase the weight on the sidekick model as the FLOPs balance improves. Figure 10 in Appendix D illustrates this behavior for Asymmetric Duos, showing the ratio between the learned temperatures of the base and sidekick models as a function of FLOPs balance. This reveals a gradual shift in weighting toward the sidekick as its relative capacity increases.
>
> ---
>
> **[Q3] Bayesian Model Averaging and Asymmetric Duos.**
> The connection between Duos and Bayesian model averaging is interesting. If one were to combine posterior samples from a large model and a small model, this could be interpreted as sampling from a hierarchical Bayesian model where the prior is a mixture over architecture types, effectively placing prior probability mass on both small and large model architectures. We agree that a Bayesian version/interpretation of Duos is an interesting avenue for future work that could make use of additional train-time or test-time computation beyond our efficient duos.
>
> ---
>
> We thank the reviewer again for the thoughtful feedback. We hope our responses adequately address the concerns raised and would be happy to discuss further during the reviewer–author discussion period.
>
> ---
>
> **References**
>
> [Opitz1999] Opitz, R. M., & Maclin, D. W. (1999). *Popular ensemble methods: An empirical study.* Journal of Artificial Intelligence Research, 11, 169–198.
>
> [Wood2023] Wood, D., Mu, T., Webb, A., Reeve, H. W. J., Luján, M., & Brown, G. (2023). *A Unified Theory of Diversity in Ensemble Learning.* Journal of Machine Learning Research, 24, 1–49.

---

> > ### Comment · Reviewer_ZFrX · 2025-08-01
> >
> > Thank you for the comprehensive rebuttal and for conducting the additional experiments comparing Asymmetric Duos with Deep Ensembles and approximate Bayesian methods. I appreciate the effort to address my concerns.
> >
> > ----
> >
> > Regarding the new comparison with Deep Ensembles ($m>2$), while the FLOPS-controlled experiment is insightful for highlighting computational efficiency, I believe it presents a somewhat unfair comparison from a practical standpoint.
> >
> > To better understand the true cost-quality trade-off, particularly for safety-critical applications, a more direct comparison is essential. Specifically, I suggest comparing your Asymmetric Duo (composed of a primary model A and a cheaper sidekick B) against a standard, high-performing Deep Ensemble built from multiple instances of that same primary model A (e.g., a 5-member ensemble of model A). If I've understood the current results correctly, this specific analysis is not yet present; please let me know if I have missed something.
> >
> > This analysis should be done without the FLOPS correction. The key question for a practitioner in a field like medical imaging is: "How much UQ performance am I giving up by choosing the efficient Duo instead of the most robust (but expensive) ensemble?" If the large Deep Ensemble provides a significantly better assessment of uncertainty, one might still choose it despite the cost.
> >
> > A brief discussion of this specific trade-off in the revised manuscript would be very helpful, as it would allow readers to properly weigh the Duo's efficiency against the absolute UQ performance required for high-stakes decisions.
> >
> > ----
> >
> > Overall, your response has confirmed my positive view of this work. It's a strong empirical paper that provides an excellent foundation for future research. I plan to increase my score, contingent on the inclusion of the aforementioned point in the final version.

---

> ### Author Response · Authors · 2025-08-04
> **Direct Comparison of Asymmetric Duos and Deep Ensembles with Fixed Base Model**
>
> We thank the reviewer for their quick and thoughtful response. We're encouraged that our rebuttal helped reinforce the reviewer's positive assessment of our work.
>
> We also appreciate the clarification regarding the Deep Ensemble comparison. We agree that this base-model-controlled setup is a meaningful and practical complement to our FLOPs-controlled comparison. Both offer useful perspectives, and we now include results under the reviewer’s requested setting using ResNet50 (RN50) for all Ensembles members and Duos $f_{\text{large}}$.
>
> | Method     | Acc↑      | Brier↓   | NLL↓        | ECE↓      | AUROC↑    | AURC↓    |
> |:-----------|:-----------|:----------|:-------------|:-----------|:-----------|:----------|
> | RN50 (Temperature Scaled) | 79.82±0.51 | 2.90±0.05 | 84.06±3.38  | 3.39±1.39  | 86.62±0.49 | 5.72±0.38 |
> | RN50+MN.75 AD (FB=0.05) | 79.88±0.42 | 2.87±0.05 | 80.45±2.17 | 3.25±0.61 | 87.11±0.29 | 5.42±0.19 |
> | RN50+SN2.0x AD (FB=0.14) | 80.32±0.32 | 2.81±0.04 | 78.93±1.78 | 2.62±0.76 | 87.00±0.27 | 5.34±0.17 |
> | RN50+ENB2 AD (FB=0.27) | 81.87±0.19 | 2.60±0.03 | 71.47±0.84 | 2.80±0.48 | 87.15±0.26 | 4.80±0.17 |
> | RN50 DEs (m=2)   | 81.12±0.29 | 2.72±0.04 | 77.12±1.90 | 3.54±1.33 | 87.19±0.30 | 4.99±0.24 |
> |...| | | | | | |
> | RN50 DEs (m=5)   | 81.97±0.13 | 2.61±0.01 | 73.16±0.90 | 3.79±0.76 | 87.62±0.11 | 4.48±0.09 |
>
> Here, Deep Ensembles follow the “pool-then-calibrate” strategy from [Rahaman2021]. Since our temperature-based Duo aggregation inherently performs calibration, this enables a fairer comparison on NLL, Brier, and ECE. All ensembles are drawn from random subsets of 10 independently trained ResNet50 (RN50) checkpoints (from both timm and torchvision). As requested, Duos in this setting are $f_{\text{large}}$-controlled: each Duo shares a fixed RN50 as $f_{\text{large}}$, and varies its sidekick among MNASNet0.75 (MN.75), ShuffleNet V2 2.0× (SN2.0x), and EfficientNet-B2 (ENB2), with FLOPs Balances (FB) of 0.05, 0.14, and 0.27, respectively.
>
> Despite their lower compute, these asymmetric RN50 Duos quickly approach performance of larger, homogeneous Deep Ensembles (e.g., $m=5$ RN50s) across most metrics. We attribute this to architectural diversity from heterogeneity [Gontijo-Lopes2022] and Duos' simple but effective temperature-based aggregation. Together, these allow RN50-based Duos to realize a large portion of the performance gains of full $m=5$ Deep Ensembles at a fraction of the cost.
>
> We hope this base-model-controlled comparison directly addresses the reviewer’s concern. We have updated the manuscript to include both the FLOPs-controlled and $f_{\text{large}}$-controlled Duo comparisons in a new Appendix section titled “Evaluating Asymmetric Duos against Deep Ensembles”. There, we note that ensembling remains a strong approach in uncertainty-critical scenarios, and Duos offer an alternative that retains much of the accuracy and UQ benefits. Relevant pointers have been added to both the main Experiments and Discussion sections.
>
> We are including the same response for Reviewer VDfk, who raised a similar concern. We appreciate this alignment, as it clearly signals the value of including this comparison in our paper. We believe this addition makes our analysis more thorough, and thank the reviewer for this thoughtful suggestion on direct comparisons between DEs and $f_{\text{large}}$-controlled ADs.
>
>
>
>
> ---
>
> **References**
>
> [Rahaman2021] Rahul Rahaman and Alexandre H Thiery. *Uncertainty quantification and deep ensembles.* Proceedings of the 35th International Conference on Neural Information Processing Systems,
> 2021.
>
> [Gontijo-Lopes2022] Raphael Gontijo-Lopes, Yann Dauphin, and Ekin Dogus Cubuk. *No one representation to rule them all: Overlapping features of training methods.* International Conference on Learning Representations, 2022.

---

> > ### Comment · Reviewer_ZFrX · 2025-08-05
> >
> > **Thank you** for providing these not-surprising, yet reassuring results in this short time frame! This is highly appreciated. I fully agree with reviewer VDfk that this makes the experimental evaluation more complete. Reflecting on all your answers, I am convinced that you addressed most of my concerns in a satisfactory manner. As a consequence, I plan on increasing my score and urge the authors to include the productive discussions and new results in the final manuscript.

---

### Official Review · Reviewer_x6b8 · 2025-07-01

**Clarity:** 3
**Significance:** 3
**Originality:** 3
**Rating:** 4
**Confidence:** 3

**Summary:**

The paper presents a combining strategy for a set of two learning models. The althors argue that the proposed model improve upon single models.

**Questions:**

- How does the method compare to dynamic or adaptive combination strategies commonly used in ensemble models?
- How does the proposed method differ from a weighting scheme with weights adjusted via cross validation
- The experiments do not compared the proposed model to state of the art models for the classsification tasks

**Ethical Concerns:**

["NO or VERY MINOR ethics concerns only"]

**Final Justification:**

Due to the additional discussions and analyses presented during the revision stage, I have decided to update the scores assigned to the paper. The contribution is now clearer, and an extensive comparison with other relevant methods has been added

**Limitations:**

The proposal is simple and thus requires an extesive experimental evaluation to support its claims. It that sense, the experimental procedure should be improved by compraing the proposed strategy to state of the art models.

**Paper Formatting Concerns:**

no formatting concerns

**Quality:**

2

**Strengths And Weaknesses:**

The proposed method is simples but the paper lacks a more robust analysis (theorectical and/or computational) to ensure that the proposed combination scheme presents any advantage when compared to other popular combination strategies

---

> ### Author Rebuttal · Authors · 2025-07-30
>
> # Author Response to Reviewer x6b8
>
> We thank the reviewer for the review.
>
> ---
>
> **\[Q1 & Q2] Aggregation Strategy Confusion**
>
> There may be some confusion about our core contribution. While our method uses an aggregation approach that is indeed standard validation-based weighting, our core contribution is applying this standard technique to models of vastly different accuracies to achieve the performance improvements of ensembles at fractional computational costs. This finding is surprising given that practical wisdom and prior work [Opitz1999; Wood2023] suggest asymmetric model combinations should not work effectively. We have added this clarification to Section 6 (Limitations and Future Directions) to better highlight our contribution, and we hope this resolves the confusion regarding the aggregation strategy.
>
> This work serves as an initial step toward understanding the aggregation strategies of deep networks of different sizes and capacities. While we focused on a small set of interpretable aggregation strategies for both point prediction and uncertainty quantification, there remains a large space of unexplored methods. Notably, our finding that simple validation-based weighting can yield effective ensembles from models with vastly different accuracies is itself a surprising and underexplored result.
>
> ---
>
> **\[Q3] Asymmetric Duos compared to SOTA methods**
>
> We are not sure what SOTA methods the reviewer is referring to.
> We built Asymmetric Duos directly off of popular and recent models that have been empirically demonstrated to be consistently best-performing through extensive evaluations by [Goldblum2023] on a suite of computer vision tasks.
> We additionally compare and combine Asymmetric Duos to recent SOTA methods like model soup [Wortsman2022] and show compatibility in Section 5.5.
> We would be happy to receive more specific feedback on which methods the reviewer is interested in seeing us benchmark against.
>
> ---
>
> **References**
>
> [Goldblum2023] Micah Goldblum, Hossein Souri, Renkun Ni, Manli Shu, Viraj Prabhu, Gowthami Somepalli, Prithvijit Chattopadhyay, Mark Ibrahim, Adrien Bardes, Judy Hoffman, et al. *Battle of the backbones: A large-scale comparison of pretrained models across computer vision tasks*. *Advances in Neural Information Processing Systems*, 36:29343–29371, 2023.
>
> [Opitz1999] Opitz, R. M., & Maclin, D. W. (1999). *Popular ensemble methods: An empirical study.* Journal of Artificial Intelligence Research, 11, 169–198.
>
> [Wortsman2022] Mitchell Wortsman, Gabriel Ilharco, Samir Ya Gadre, Rebecca Roelofs, Raphael Gontijo-Lopes, Ari S. Morcos, Hongseok Namkoong, Ali Farhadi, Yair Carmon, Simon Kornblith, *et al.* *Model soups: Averaging weights of multiple fine-tuned models improves accuracy without increasing inference time.* In *International Conference on Machine Learning*, pages 23965–23998. PMLR, 2022.
>
> [Wood2023] Wood, D., Mu, T., Webb, A., Reeve, H. W. J., Luján, M., & Brown, G. (2023). *A Unified Theory of Diversity in Ensemble Learning.* Journal of Machine Learning Research, 24, 1–49.

---

> > ### Comment · Reviewer_x6b8 · 2025-08-04
> >
> > Thank you for the detailed rebuttal and for clearing up those misunderstandings, The additional experiments were a valuable contribution.
> >
> > Regarding the comparison to SOTA methods, it is interesting to see how Asymmetric Duos compare to ensemble of deep models.

---

> > > ### Author Response · Authors · 2025-08-04
> > > **Direct Comparison of Asymmetric Duos and Deep Ensembles with Fixed Base Model**
> > >
> > > We thank the reviewer for the response.
> > >
> > > Here we present the requested direct comparison of Deep Ensembles and Asymmetric Duos, using ResNet50 (RN50) for ensemble members and Duo $f_{\text{large}}$.
> > >
> > > | Method     | Acc↑      | Brier↓   | NLL↓        | ECE↓      | AUROC↑    | AURC↓    |
> > > |:-----------|:-----------|:----------|:-------------|:-----------|:-----------|:----------|
> > > | RN50 (Temperature Scaled) | 79.82±0.51 | 2.90±0.05 | 84.06±3.38  | 3.39±1.39  | 86.62±0.49 | 5.72±0.38 |
> > > | RN50+MN.75 AD (FB=0.05) | 79.88±0.42 | 2.87±0.05 | 80.45±2.17 | 3.25±0.61 | 87.11±0.29 | 5.42±0.19 |
> > > | RN50+SN2.0x AD (FB=0.14) | 80.32±0.32 | 2.81±0.04 | 78.93±1.78 | 2.62±0.76 | 87.00±0.27 | 5.34±0.17 |
> > > | RN50+ENB2 AD (FB=0.27) | 81.87±0.19 | 2.60±0.03 | 71.47±0.84 | 2.80±0.48 | 87.15±0.26 | 4.80±0.17 |
> > > | RN50 DEs (m=2)   | 81.12±0.29 | 2.72±0.04 | 77.12±1.90 | 3.54±1.33 | 87.19±0.30 | 4.99±0.24 |
> > > |...| | | | | | |
> > > | RN50 DEs (m=5)   | 81.97±0.13 | 2.61±0.01 | 73.16±0.90 | 3.79±0.76 | 87.62±0.11 | 4.48±0.09 |
> > >
> > > Here, Deep Ensembles follow the “pool-then-calibrate” strategy from [Rahaman2021]. Since our temperature-based Duo aggregation inherently performs calibration, this enables a fairer comparison on NLL, Brier, and ECE. All ensembles are drawn from random subsets of 10 independently trained ResNet50 (RN50) checkpoints (from both timm and torchvision). Duos in this setting are $f_{\text{large}}$-controlled: each Duo used a RN50 as $f_{\text{large}}$, and varies its sidekick among MNASNet0.75 (MN.75), ShuffleNet V2 2.0× (SN2.0x), and EfficientNet-B2 (ENB2), with FLOPs Balances (FB) of 0.05, 0.14, and 0.27, respectively.
> > >
> > > Despite their lower compute, these asymmetric RN50 Duos quickly approach performance of larger, homogeneous Deep Ensembles (e.g., $m=5$ RN50s) across most metrics. We attribute this to architectural diversity from heterogeneity [Gontijo-Lopes2022] and Duos' simple but effective temperature-based aggregation. Together, these allow RN50-based Duos to realize a large portion of the performance gains of full $m=5$ Deep Ensembles at a fraction of the cost.
> > >
> > > We hope this base-model-controlled comparison directly addresses the reviewer’s concern. We have updated the manuscript to include both the FLOPs-controlled and $f_{\text{large}}$-controlled Duo comparisons in a new Appendix section titled “Evaluating Asymmetric Duos against Deep Ensembles”. There, we note that ensembling remains a strong approach in uncertainty-critical scenarios, while Duos offer a cheaper alternative that retains much of the accuracy and UQ benefits. Relevant pointers have been added to both the main Experiments and Discussion sections.
> > >
> > > Note that we are including the same analysis result for Reviewer ZFrX and VDfk, who raised a similar concern. We believe this addition makes our analysis more thorough and helps better position Duos among other methods for practitioners, and thank the reviewer for this thoughtful suggestion on direct comparisons between DEs and ADs. If this additional comparison addresses your outstanding issues, we'd appreciate if you adjust your score.
> > >
> > > ---
> > >
> > > **References**
> > >
> > > [Rahaman2021] Rahul Rahaman and Alexandre H Thiery. *Uncertainty quantification and deep ensembles.* Proceedings of the 35th International Conference on Neural Information Processing Systems,
> > > 2021.
> > >
> > > [Gontijo-Lopes2022] Raphael Gontijo-Lopes, Yann Dauphin, and Ekin Dogus Cubuk. *No one representation to rule them all: Overlapping features of training methods.* International Conference on Learning Representations, 2022.

---

### Official Review · Reviewer_72FX · 2025-07-02

**Clarity:** 3
**Significance:** 3
**Originality:** 2
**Rating:** 5
**Confidence:** 4

**Summary:**

The paper presents "Asymmetric Duo," a novel efficiency-focused approach to uncertainty estimation that challenges conventional ensembling methods. Rather than the computationally prohibitive practice of training multiple large models, the researchers demonstrate how pairing a heavyweight primary model with a lightweight companion model yields remarkable benefits. By implementing a learned weighting mechanism to blend predictions from these mismatched architectures (such as combining a ViT-B with a much smaller ResNet-34), the system achieves enhanced performance across multiple dimensions. Testing across five image classification datasets revealed that this asymmetric pairing not only preserves but actually improves the primary model's accuracy and uncertainty calibration while requiring only fractional additional computational resources. Particularly noteworthy is that this performance boost occurs despite the significant capability gap between the models, working effectively even when applied to sophisticated techniques like model soups. This represents a practical breakthrough for deploying uncertainty-aware deep learning in resource-constrained environments.

**Questions:**

- Are the large and small models trained on the same set of data? If so, what would happen if the training data is not available to the smaller model?
- The paper only considered a duo. Does adding more small models further improve the performance?

**Ethical Concerns:**

["NO or VERY MINOR ethics concerns only"]

**Final Justification:**

I have read the authors rebuttal and also the other reviewers' feedback. Despite several weaknesses, I think the findings of the paper is interesting. I am inclined towards accepting the paper.

**Limitations:**

Yes.

**Paper Formatting Concerns:**

No.

**Quality:**

3

**Strengths And Weaknesses:**

## Strengths:
- The paper tackles an important problem.
- The idea of combining a large model is simple yet smart.
- The results that combining a large model prediction with a small one can lead to enhanced classification accuracy is pleasantly surprising to me.
- Extensive experiments were conducted to validate the claims.

## Weaknesses:
- Though empirically supported, the proposed method is not backed up by any theoretical foundations. It would be better if there are interesting theoretical insights to further support the proposed method.
- Although most of the Duos lead to performance improvements, the relative improvements can vary given the same FLOPs Balance. It is not immediately clear to me on how we can select the optimal model/configuration to pair up with the large model.

---

> ### Author Rebuttal · Authors · 2025-07-30
>
> # Author Response to Reviewer 72FX
>
> We thank the reviewer for the thoughtful review and insightful questions.
>
> ---
>
> **[W1] Regarding theoretical foundations**
> We appreciate this feedback. While we focused on empirical validation given the practical nature of our contribution, we could easily establish several straightforward theoretical results.
>
> *Accuracy improvement guarantee*
> If we select temperature weights to maximize validation accuracy, Duos necessarily outperform the large model on validation data (the large model is simply a Duo with weights $[1,0]$). Given the low VC dimension of the space of possible temperature weightings ($\leq 3$), standard learning theory provides $O\left(\sqrt{1/|\mathrm{val\_set}|}\right)$ generalization bounds on the test accuracy improvement.
>
> *Ensemble equivalence*
> Recent work [Dern2025, Abe2022] show large models behave like ensembles of smaller models. A direct application of Theorem 3.3 from Dern2025 proves that a Duo combining M×D and D-parameter random feature models is equivalent to an ensemble of $(M+1)$ D-parameter models—formalizing our intuition about why asymmetric combinations work.
>
> Both proofs are approximately 10 lines and follow from standard techniques. We omitted them to avoid cluttering the paper, but we're happy to include whichever result you feel would most benefit readers. Would either theorem meaningfully strengthen the contribution in your view?
>
> ---
>
> **[W2] Practical Selection of Duos**
> We thank the reviewer for this thoughtful point regarding the practicality of selecting the sidekick model in an Asymmetric Duo, a question that interests us as well.
>
> Similar to choosing a backbone model for fine-tuning, it is indeed difficult to know beforehand which Duo will perform best on a specific downstream task. That said, some heuristic indicators have been proven useful. In particular, we have found that FLOPs balance and pre-training performance on a standard dataset like ImageNet are informative when choosing the sidekick model. FLOPs balance is free and reliable, and is aligned with the general "larger models perform better" intuition in modern ML. Lastly, as the review suggested, *most* Duos lead to performance improvements, so even though it's hard to predict the optimal choice of sidekick, it is easy to find good contenders.
>
> ---
>
> **[Q1] Training Dataset**
> Yes, both the base model and the sidekick model in our experiments have been trained on the same downstream training set, though this is not mandatory. We can still make Duos even if a sidekick model has only seen partial or none of the entire training set, and this would be an interesting investigation for future work.
>
> ---
>
> **[Q2] Adding more than one sidekick model**
> We thank the reviewer for pointing out this natural follow-up of our Duo framework.
>
> Combining a large base model with more than one sidekick can be considered as another approach to scale up the Asymmetric framework, and its efficiency/performance tradeoff is one that we find interesting as well. We conducted additional experiments using a Trio setup, aggregating a base model with two sidekicks using the same temperature weighting strategy described in Section 3.1, while keeping the total additional FLOPs within 20%. While Trios sometimes show slight improvements over Duos in AUROC and AURC, we also observe some regressions in Accuracy, Brier score, and NLL.
>
> Overall, we find that Asymmetric Duos offer a more consistent and efficient tradeoff at low FLOPs budgets, making them the more practical choice in most settings.
>
> ---
>
> **References**
>
> [Abe2022] Abe, Buchanan, Pleiss, Zemel, Cunningham. *Deep Ensembles Work, But Are They Necessary?* NeurIPS 2022.
>
> [Dern2025] Niclas Dern, John P. Cunningham, and Geoff Pleiss. *Theoretical limitations of ensembles in the age of overparameterization* ICML 2025.

---

> > ### Comment · Reviewer_72FX · 2025-08-05
> >
> > Thanks a lot for addressing my concerns. I recommend incorporating some of these comments to the paper.

---

### Note · Authors · 2025-08-13

As the Author-Reviewer discussions period ends, we would like to thank the reviewers for their active engagement during the discussions. We believe comparisons between Asymmetric Duos and Deep Ensembles (ZFrX, VDfk, x6b8), comparisons against scalable Bayesian methods (ZFrX), analyses of logits norms and temperature behaviors (VDfk, ZFrX), and discussions on duo practicality (72FX) and scalability to Asymmetric Trios (72FX), amongst other reviewer suggestions, have meaningfully strengthened our work. We appreciate these insightful suggestions and the productive, pleasant rebuttal and discussion experience.

---

### Decision · Program_Chairs · 2025-09-17

**Decision:**

Accept (spotlight)

**Comment:**

This well-written and well-motivated paper proposes a new approach to uncertainty estimation of large-scale models that is computationally savvy. It has been evaluated by 4 knowledgeable reviewers who all agreed that it is acceptable for NeurIPS (3 straight accept scores, one marginal accept), despite several noted weaknesses (some of which the authors have managed to address in the rebuttal and discussion with the reviewers). The authors are strongly encouraged to carefully reflect on the feedback received in the final revision of the paper.